



# Laser Ablation - ICP-MS measurements for high resolution chemical ice core analyses with a first application to an ice core from Skytrain Ice Rise (Antarctica)

Helene Hoffmann[1,4], Jason Day[1], Rachael Rhodes[1], Mackenzie Grieman[1,5], Jack Humby[2], Isobel Rowell[1], Christoph Nehrbass-Ahles[1,3], Robert Mulvaney[2], Sally Gibson[1], and Eric Wolff[1]

[1]University of Cambridge, Department of Earth Sciences, Downing Street, Cambridge CB2 3EQ, UK
[2]British Antarctic Survey, High Cross, Madingley Road, Cambridge CB3 0ET, UK
[3]National Physical Laboratory, Hampton Road, Teddington TW11 0LW, UK
[4]University of Tübingen, Department of Geoscience, Schnarrenbergstr. 94-96, 72076 Tübingen, Germany
[5]Springer Nature America, c/o WeWork, 1100 15th Street, N.W. Floor 04-W184, Washington DC, 20005, USA

**Correspondence:** Helene Hoffmann (he.hoffmann@uni-tuebingen.de)

**Abstract.** For many ice core related scientific questions, like the analysis of fast changing climate signals, the depth resolution of conventional methods of analysis is not sufficient. In this study we present a setup of Laser Ablation Inductively Coupled Plasma - Mass Spectrometry (LA-ICP-MS) for high resolution glacier ice impurity analysis to the sub-millimetre scale. This setup enables chemical impurity ice core analysis to a depth resolution of $\sim 80\,\mu m$ while consuming only minimal amounts
of ice. The system was used for simultaneous analysis of sodium, magnesium and aluminium incorporated in the ice structure. In a first case study within the framework of the WACSWAIN (WArm Climate Stability of the West Antarctic ice sheet in the last INterglacial) project, the method was applied to a selection of samples from Skytrain ice core (West Antarctica), on a total length of 6.7 m consisting of about 130 single samples. The main goal of this study was to use the new LA-ICP-MS method to extract meaningful climate signals on a depth resolution level beyond the limits of Continuous Flow Analysis (CFA).
A comparison between low resolution CFA data and the high resolution LA-ICP-MS data reveals generally good agreement on the decimetre scale. Stacking of parallel laser measurements together with frequency analysis is used to analyse the high resolution LA-ICP-MS data on the millimetre scale. Spectral analysis revealed that despite effects of impurity accumulation in ice crystal grain boundaries, periodic concentration changes in the Skytrain ice core on the millimetre scale can be identified in ice from 26 ka BP (kilo years before present; 1950 CE) and also from the Last Interglacial (LIG). These findings can open
new possibilities for climate data interpretation with respect to fast changes in the last glacial and beyond.

## 1 Introduction

Ice cores from polar and non-polar regions are invaluable archives of past atmospheric conditions and climate information. Chemical and physical analyses of the internal ice core layers which were formed by sequential compaction of snow to ice are the key to deciphering this information. With increasing depth, pressure and due to the flow of the glacier ice, (annual) layers of
varying deposited impurity concentration can become highly thinned to the sub-millimeter level (Faria et al., 2010). Techniques



therefore need to be developed to investigate climatic changes exhibited in ice core records at high depth resolutions. The aim of this study is to present the new Cambridge LA-ICP-MS ice measurement system and assess its capabilities to resolve fine scale structures and layers in deep sections of Skytrain ice core. This will ultimately enable interpretation of climate-related signals in deep and highly thinned ice from the core, which was drilled as part of the WArm Climate Stability of the West Antarctic ice

sheet in the last INterglacial (WACSWAIN), ERC (2017) project. We define layering as periodic changes in concentration of the respective impurities with depth, which is superimposed onto concentration changes caused by effects in the ice microstructure. These changes can manifest on time scales from seasonal and sub-annual to multi-millenial. The analysis is focused on the major elements associated with marine and terrestrial influences: Na, Mg, Al and Ca. The aim of the WACSWAIN project is to decipher the fate of the Filchner-Ronne Ice Shelf during the last interglacial (LIG) ∼ 110 -130 ka before present (year 1950

CE). This information is relevant to estimating future sea level changes under a warming climate. As part of this project, the Skytrain ice core was drilled at Skytrain Ice Rise (Mulvaney et al., 2021) on the edge of the Filchner-Ronne Ice Shelf (Fig. 1). If the ice sheet partly or entirely collapsed during the LIG, but Skytrain Ice Rise remained ice covered, it would be more marine-influenced than with an intact ice sheet offshore. This enhanced marine influence would appear as an increase in sea salt concentrations in the ice core record. Ice from the LIG was detected in the ice core at 605-631 m depth, encompassing a

time range of about 108 - 126 ka BP. The estimated annual layer thickness from the age model in this depth section is on the order of 1.5 mm (Mulvaney et al., 2022). Therefore, techniques that can detect sea salt and other impurity variability at very high depth resolution are needed to investigate fast changes in the ice dynamics near Skytrain Ice Rise during the LIG.

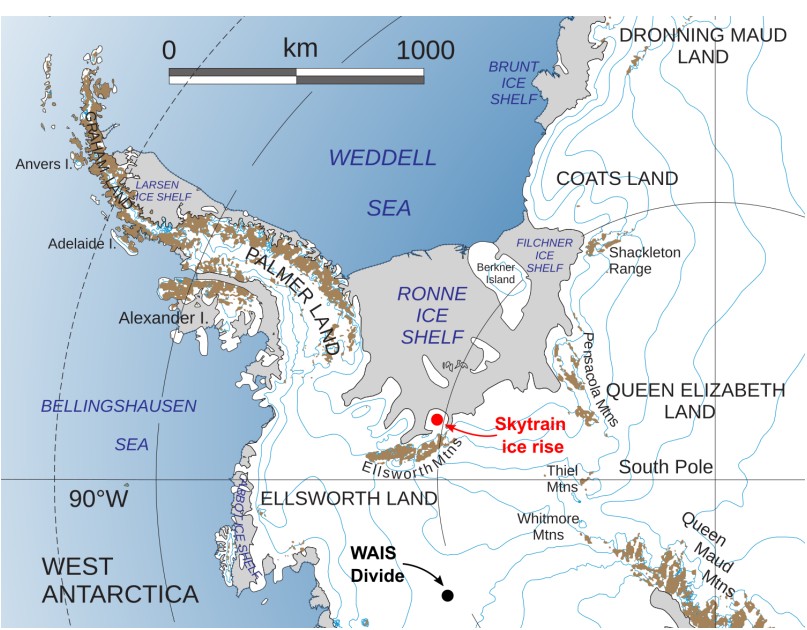

**Figure 1.** Location of Skytrain Ice Rise at the edge of the Ronne Ice Shelf. Adapted from overview map, courtesy of USGS (2022)





In past studies, chemical impurity analyses of ice cores were mainly carried out by discrete sampling of the core material followed by ion chromatography (Legrand and Mayewski, 1997; Littot et al., 2002). Although a depth resolution in the lower centimeter range can be achieved by these methods, they are time-consuming, labour-intensive and not applicable for high resolution analysis of long ice core records. The development of so called Continuous-Flow-Analysis (CFA) techniques massively improved the capabilities of chemical ice core analysis mainly in analysis speed, sample handling and also depth resolution (Röthlisberger et al., 2000; McConnell et al., 2002; Cole-Dai et al., 2006; Bigler et al., 2011). However, these continuous methods are still limited to depth resolutions of at best about 1 cm (Bigler et al., 2011). This is not high enough to address many scientific questions involving very old, highly thinned ice close to bedrock and ice microstructure. A project where this becomes highly relevant is the current search for 1-1.5 million year old ice in Antarctica in projects like COLDEX and Beyond EPICA (Parrenin et al., 2017; Brook et al., 2022) . In recent years, the technique of LA-ICP-MS was adapted for glacier ice analysis (Reinhardt et al., 2003; Müller et al., 2011; Sneed et al., 2015; Bohleber et al., 2020). This technique enables virtually non-destructive impurity measurements of ice samples to a depth resolution on the order of ∼ 100 µm or even below. The main principle of analysis is the same for all LA-ICP-MS systems. A highly energetic (UV) laser beam slowly scans the ice sample surface and ablates small amounts of material, which are subsequently purged via a stream of carrier gas into an ICP-MS for elemental analysis. The laser ablation systems currently in operation differ mainly in the design of the cryocell which is used to accommodate the glacier ice samples during ablation. Most of the setups are designed to hold small strips of ice with lateral extensions up to ∼ 10 cm maximum (Bohleber et al., 2020) in a closed laser cell. To our knowledge at present only the system at the University of Maine is built to analyse half core pieces of up to 1 m length (Sneed et al., 2015).This system involves an open cooling chamber with a small laser cell that is attached to the sample surface and incrementally moved along the depth axis of the sample.

The Cambridge LA-ICP-MS system is routinely used for geological investigations e.g. Jackson and Gibson (2018). In this study, it was modified to enable glacier ice analysis. The Cambridge University setup differs from the University of Maine system in that it is a closed cell design, which is able to hold sample geometries in the shape of standard microscope slides (75 x 26 mm). It is therefore designed for detailed impurity analysis rather than for large sample throughput. In this study, the capability of the new Cambridge LA-ICP-MS ice measurement system to resolve fine scale structures and (annual) Na, Mg, Al, and Ca concentration changes in deep sections of Skytrain ice core is assessed.

## 1.1 WACSWAIN project and Skytrain ice core

Within the framework of the WACSWAIN project, the Skytrain ice core was drilled to bedrock at Skytrain Ice Rise (see Fig. 1) in field season 2018/19 in a joint effort between the British Antarctic Survey and the University of Cambridge (Mulvaney et al., 2021). The drill reached bedrock at 651 m total depth and with a basal temperature of - 15 °C; the basal ice showed no signs of melting (Mulvaney et al., 2021). The surface snow accumulation rate was determined to be about 13.5 cm water equivalent per year (w.e./yr) (Hoffmann et al., 2022). The whole core length was analysed via CFA at the British Antarctic Survey (Grieman et al., 2021; Hoffmann et al., 2022). An overview of the previous analytical methods used and the respective depth resolutions of the chemical species relevant for this study is given in Table 1. Dating of the ice core was completed via



a combined approach of annual layer counting, identification of absolute age markers (e.g. volcanic eruptions), matching of features (e.g. in the methane profile) to dated cores and flow modelling (Hoffmann et al., 2022; Mulvaney et al., 2022).

**Table 1.** Overview of selected analytes measured during the continuous flow analysis of Skytrain ice core (Hoffmann et al., 2022). FIC stands for Fast Ion Chromatography. The term Depth resolution refers to the moving average intervals for the respective method. For the FIC, this Depth resolution is a binned average (Grieman et al., 2021). The most important parameters for this study are highlighted in **bold**.

| Instrument type | Model | Analytes | Depth resolution |
|---|---|---|---|
| ICP-MS | Agilent 7700x | $^{23}$**Na**, $^{24}$**Mg**, $^{27}$**Al**, $^{43}$**Ca**, $^{44}$**Ca** | 3.8 - 4.7 cm |
| FIC cation | Dionex ICS-3000 | **Na$^+$, Ca$^{2+}$, Mg$^{2+}$**, K$^+$ | > 4cm |
| Fluorescence | FIALab Precision Measurement Technologies photomultiplier tube fluorometers (PMT-FL) | **Ca$^{2+}$**, NH$_4^+$, H$_2$O$_2$ | 1.4 cm |

## 2 Methods

### 2.1 Sample preparation and handling

In this study, a total length of 6.7 m of Skytrain ice core consisting of about 130 single samples was analysed. All ice samples were prepared in the cold lab facilities of the British Antarctic Survey (Cambridge) at -25°C. Each sample was initially cut with a band saw to a geometry of exactly 5 cm in length (along the depth axis), about 2 cm in width and about 2 cm in thickness, depending on the ice core cut that was used. For Skytrain, the samples were taken from the cut for physical properties analyses. It is directly adjacent to the CFA piece (see cutting plan in Grieman et al. (2021)). The Physical Properties and CFA piece share a common surface and the analysed ice core sections are horizontally only separated by about 2 cm. Thus, the same signals are expected in both the CFA and the LA-ICP-MS datasets. After the first cut, the largest surface of the sample is cut down by 2 mm and smoothed using a sledge microtome (Bright 8000) and ceramic coated blades. This polished ice surface is then mounted with a line of ultra pure water (ELGA Labwater, 18 MΩ cm, commonly referred to as MQ-water) around the edges onto a pre-cleaned (using isopropyl alcohol) standard microscope slide. Quartz slides were used for liquid nitrogen (LN) storage to avoid splintering of the ice due to thermal expansion effects. Subsequently, the samples on the slides were cut a second time with the band saw to a thickness of about 5 mm. The samples were then polished with the microtome to a thickness of < 3 mm including the slide (1 mm). Photos were taken of all samples before storing them (see Fig. 2) in slide mailers either at -25°C for short term or in liquid nitrogen dry shippers for long term storage of more than three days to minimise sublimation. The microtome blade was thoroughly cleaned with a brush after every preparation step and changed on a daily basis. All materials





in direct contact with the ice samples were metal free. On the day of analysis, the samples were transferred in an insulated container to the UNiversity of Cambridge and stored at -18°C until measurement.

## 2.2 Setup and characteristics of the Cambridge LA-ICP-MS system

The Cambridge LA-ICP-MS system comprises an ESI UP193 Ultra Compact laser, which operates at 193 nm (NWR193) and is coupled to a Perkin Elmer Nexion 350D quadrupole ICP-MS. During analysis, the ICP-MS was operated without use of the collision cell because operation of the collision cell caused a signal loss that outweighed any benefit of background suppression for our glacier ice application.

**Table 2.** Specifications and typical settings of the Cambridge LA-ICP-MS system for analysis of glacier ice samples.

| **Perkin Elmer® Nexion 350D ICP-MS Parameter settings** | |
| --- | --- |
| Dwell times per mass [ms] | 50-100 for $^{23}$Na |
| | 100 for $^{24}$Mg, $^{27}$Al |
| | 200 for $^{43}$Ca |
| RF power | 1500 W |
| Plasma gas flow | 18 L min$^{-1}$ |
| Auxiliary gas flow | 1.2 L min$^{-1}$ |
| Nebulizer gas flow | 0.84 - 1.10 L min$^{-1}$ optimized daily |
| ThO | < 1%, typically $\sim$ 0.2 % |
| **NWR193 Laser Ablation system Parameter** | |
| Laser and wavelength | Coherent ExciStar XS excimer laser 193 nm |
| Helium gas flow | 600-800 mL min$^{-1}$ |
| Laser spot size | round: 150 μm diameter |
| | rectangular: 100-120 μm x-direction, |
| | 50 μm y-direction |
| Laser repetition rate | 20-120 Hz depending on sample |
| Laser fluence on target | round spot: 6 J cm$^{-2}$ |
| | rectangular spot: 36-50 J cm$^{-2}$ |
| Cryocell temperature | -15°C to -18°C Peltier cooling |

Prior to analysis, the ICP-MS was cleaned with diluted HNO$_3$ and MQ water in solution mode to flush out any residual contamination in the instrument. The daily tuning routine included automated optimisation of the QID (Quadrupole Ion Deflector) and the nebulizer gas flow. At the beginning of each analysis, which typically lasted 3-4 days comprising $\sim$ 20-30 samples, the torch position was also optimised. The ICP-MS calibration and tuning routines were performed using NIST 612 reference material (Jochum et al., 2011) in the standard laser cell, not the cryocell. An overview of the typical parameters of the ICP-MS





and the laser system can be found in Table 2. The laser ablation system is equipped with a 100 x 100 mm two volume sample chamber including an inner cup, which enables high precision and short purging and washout times (see Fig. 3).

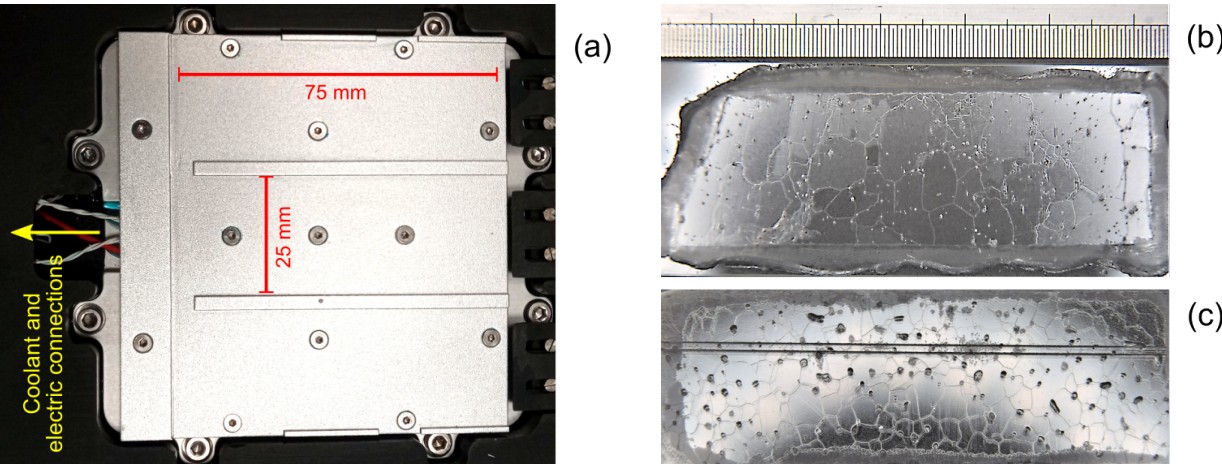

**Figure 2.** (a): Top view of the cryostage. There are three slots available to insert standard microscope slides. The aluminium surface is cooled via Peltier elements. (b): Sample from Skytrain ice core (depth: 615.22-615.27 m, LIG ice) on microscope slide before lasering. The scale is in cm. The bubbles and crystal grain boundaries are clearly visible. Top of core piece to the left. (c): Sample from Skytrain ice core (83.20-83.25 m, ca. 550 years before present) two parallel laser paths spaced by 1 mm are visible, spot size was 150 μm. Top of core piece to the left.

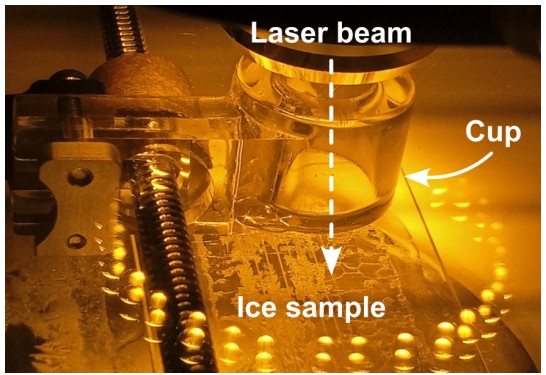

**Figure 3.** View into the laser chamber during glacier ice analysis. There are several lines visible on the ice surface. The laser beam is focused onto the ice in the center of the small volume cup, which reduces mixing and enables fast washout and high measurement precision.

After the tuning process of the ICP-MS, the standard sample stage was exchanged for the CryoCell. This sample stage is designed to accommodate three samples in the geometry of standard microscope slides (see Fig. 2). It was adapted to hold samples of a total thickness of 3 mm (including the 1 mm slide). The CryoCell is cooled by a two-stage system. The pre-cooling is conducted by circulation of a coolant through the stage at a temperature of 4°C followed by cooling via Peltier



elements. Operating temperatures on the stage surface for ice samples were typically -15°C to -18°C. Set temperatures are
stable within ± 0.5°C over the course of the day. The sample cup inside the chamber is directly connected to the ICP-MS by
Tygon® tubing, which is attached to a DCI (Dual Concentric Injector). This injector enables fast washout times and minimises
turbulent mixing of sample material in the lines (Douglas et al., 2015). The whole body of the laser ablation chamber was
wrapped into a glove bag, which was purged with dry nitrogen. This means the system was enclosed entirely in polyethylene
(PE) foil and samples were handled through gloves attached to the PE bag from the outside to prevent any contact of the open
sample surface with the lab air. An open reservoir of liquid nitrogen (LN) was placed inside the bag to act as a moisture trap and
thus minimise condensation of remaining air humidity on the sample and the CryoCell window surface. Immediately before
analysis, the ice samples were repacked from a chest freezer at -18°C into a styrofoam box. The glove bag, covering the laser
chamber housing, was opened and the sample box placed quickly inside. It was sealed again and the interior of the bag was
purged with dry nitrogen. Then from the outside of the bag the laser chamber was opened, the sample stage taken out and the
glacier ice samples quickly transferred into the holding slots by use of a PTFE coated pair of tweezers. The transfer time in
which the ice samples were not actively cooled was therefore minimised to about 10-15 seconds depending on the skills of the
operator. The stage was then pushed back into the ablation cell, which was then purged with dry helium to remove any residual
ambient air and moisture. With this approach we were able to avoid condensation on the sample surface and melting of the ice.
Even after 2 hours in the cell, no condensation of humidity or significant sublimation could be observed.

### 2.3   Optimisation of laser ablation parameters

#### 2.3.1   Blank and standard samples

For reliable analysis and signal interpretation, it is essential to have a comprehensive understanding of the procedural blank.
The procedural blank of the sample preparation and laser ablation process was assessed by analysis of artificial ice samples
made from MQ water. This blank ice was produced by freezing of the water on a lab shaker at -20°C in PE bags which were
pre-cleaned with nitric acid. The slow freezing leads to concentration of the contaminants (solutes and particles) in the section
that freezes last, which can then be cut out and discarded (Shafique et al., 2012; Hoffmann et al., 2018). All blank samples
were cut from the remaining clean ice, microtomed and handled in the exact same way as the real ice samples. Creating
artificial ice samples with known impurity concentrations to serve as standard material of the same matrix to calibrate the
laser ablation process was attempted. Because of separation effects during the freezing process (Halde, 1980), it is extremely
difficult to create ice with a homogeneous impurity distribution. Repeated flash freezing of small amounts of standard solution
as described by Della Lunga et al. (2017) might be the most promising approach. However, in the course of this study it was not
possible to produce sufficiently homogeneous ice samples to serve as reference materials using this procedure. It was therefore
decided that the laser ablation process would not be calibrated in the quantitative sense. It is rather tuned exclusively with
NIST 612 reference material (glass) to optimise the ablation parameters and the signal quality regarding signal intensity and
background reduction. The aim of the present study is to investigate relative changes in the retrieved laser ablation signals



and not to determine absolute impurity concentrations. It is therefore sufficient to tune the laser using the routine NIST 612 standard.

### 2.3.2 Washout time

A crucial parameter to achieve high spatial resolution of the retrieved impurity data is a fast washout time from the laser
system to the ICP-MS. This minimises the negative effects of turbulence and mixing in the connecting tubing and thus enables clear separation of highly variable signal peaks at the μm scale. The Cambridge LA-ICP-MS system is equipped with a Dual Concentric Injector (DCI, manufactured by ESL) which enables immediate introduction of the sample into the plasma stream while minimising turbulent mixing in a nebulizer (Douglas et al., 2015). The washout time was determined by a series of 10 subsequent laser pulses on NIST 612, with a 50 μm round spot at a firing rate of 1 Hz. A comparison of the retrieved signals
with and without use of the DCI is shown in Fig. 4.

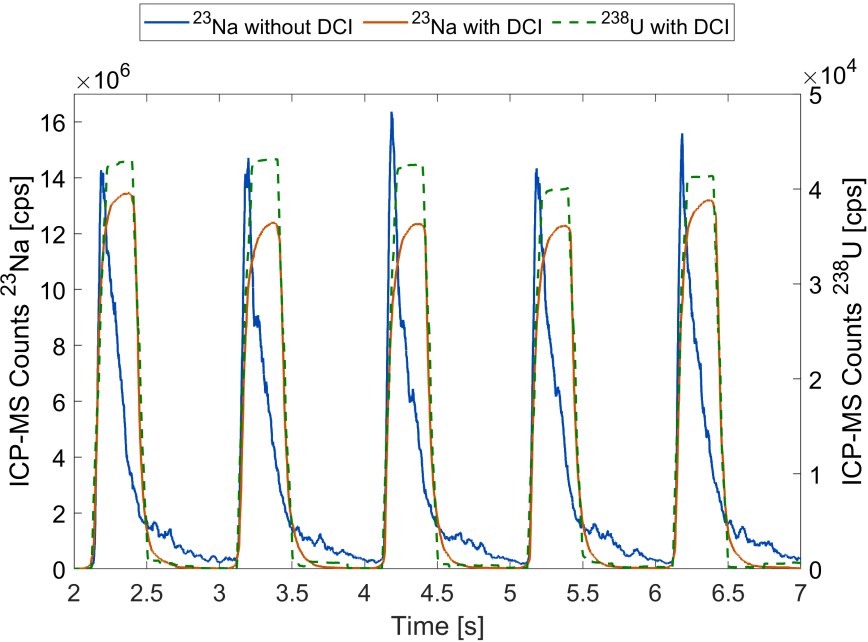

**Figure 4.** Washout times for $^{23}$Na and $^{238}$Uusing the NIST 612 standard material. The washout times using the DCI are a factor of two lower than without it. Without DCI the 1% level is not reached before the next pulse.

The washout times were calculated using the 10% and 1% full width maximum criterion. This means that the decay time of the signal from peak intensity to 10% (or 1% respectively) of the highest level is measured. The results are reported in Table 3. The use of the DCI decreased the washout time to the 10% level by a factor of two. Without the DCI, the 1% level could not be reached before the start of the next pulse.



**Table 3.** Washout times for NIST 612 glass calculated for $^{23}$Na and $^{238}$U with and without use of the DCI.

| Monitored mass | 10% full width [ms] | 1% full width [ms] |
|---|---|---|
| $^{23}$Na without DCI | $32.5 \pm 3.4$ | not reached |
| $^{23}$Na with DCI | $14.1 \pm 1.0$ | $299.1 \pm 13.0$ |
| $^{238}$U with DCI | $10.1 \pm 0.3$ | $21.1 \pm 0.1$ |

A washout time of about 300 ms to the 1% level for Na was found. This is better than results for older ice LA-ICP-MS systems (Della Lunga et al., 2014; Spaulding et al., 2017) and comparing the results for $^{238}$U (21 ms) even better than the current state-of-the-art setup (34 ms for $^{238}$U in Bohleber et al. (2020)).

### 2.3.3 Ablation settings and resolution

Laser ablation settings were optimised in each measurement session to account for the different sample and resolution require-
ments. This includes the drift of the laser system sensitvity and the sample surface conditions. For very transparent clear ice, with little impurity and bubble content, a higher repetition rate and slower scanning speed was required to achieve a reliable coupling of the laser beam with the ice surface. The masses analyzed were $^{23}$Na, $^{24}$Mg, $^{27}$Al and $^{43}$Ca in order to compare directly the results of the LA-ICP-MS method and the CFA ICP-MS method. $^{43}$Ca was chosen over $^{44}$Ca because it has less molecular interferences. An extension of the monitored masses to e. g. $^{56}$Fe by use of the collision cell only for such analysis
in the future is anticipated. ICP-MS dwell times were 50 ms or 100 ms for Na, 100 ms for Mg and Al and 200 ms for Ca amounting to a total acquisition time of 500 ms for one cycle. Therefore the acquisition time is the dominating factor regarding depth resolution compared to the much smaller washout time. Paths along the depth axis of the ice core samples were scanned, typically in the range of 4-5 cm over the whole sample length. To remove surface contamination, a fast pre-ablation scanning at speeds of 100-220 μm s$^{-1}$ and 40-100 Hz laser frequency was performed for each line. The wide range of frequencies and
scanning speeds results from the varying sample conditions (transparent ice or bubbly / opaque ice). The scanning speed for the following final ablation was usually 40 μm s$^{-1}$ at repetition rates of 20-150 Hz depending on the sample. This results in a total measurement time of $\sim$ 25 min for each line of 5 cm. For ablation, depending on the intended depth resolution either a round spot of typically 150 μm diameter for low resolution or a rectangular spot (50 μm in depth, 50-100 μm in horizontal direction) for high resolution was used. The resulting laser fluence on the sample surface was therefore in the range of 6-8 J
cm$^{-2}$ for the round spot and between 36 and 55 J cm$^{-2}$ for the rectangular spot. The resulting depth resolution, accounting for ICP-MS acquisition time, scanning speed and laser repetition rate, was calculated using an approach similar to that used in Sneed et al. (2015) and Spaulding et al. (2017). For typical low resolution settings (150 μm spot size, 20 Hz repetition rate, 40 μm s$^{-1}$, 500 ms acquisition time) this results in a resolution along the core depth of 170 μm. Taking into account the 1% washout time of $\sim$ 300 ms for Na, the laser travels 12 μm during that time, which needs to be added. The total depth resolution
for this setting is then 182 μm. For a typical high resolution setting (50 μm spot size, 150 Hz repetition rate, 40 μm s$^{-1}$) this is reduced to 81.5 μm. The achievable depth resolution with the Cambridge system is therefore at least one order of magnitude





higher than the expected annual layer thickness in the LIG section of the Skytrain ice core (0.8 - 1 mm). Therefore it should be possible to resolve layered structures on the order of the annual layer thickness in the ice core, if these structures are physically and chemically preserved.

## 3 Results and Discussion

### 3.1 Background determination

The procedural blank for each element was determined by analysis of artificial MQ ice (see section 2.3.1). Some results of these measurements for $^{23}$Na are shown in Fig. 5. The MQ samples were handled and pre-ablated like the ice core samples, and the laser settings for the lowest depth resolution were used. For the first 20 and the last 10 seconds of the measurement (grey areas in Fig. 5), the ICP-MS recorded the gas blank of helium without any laser signal.

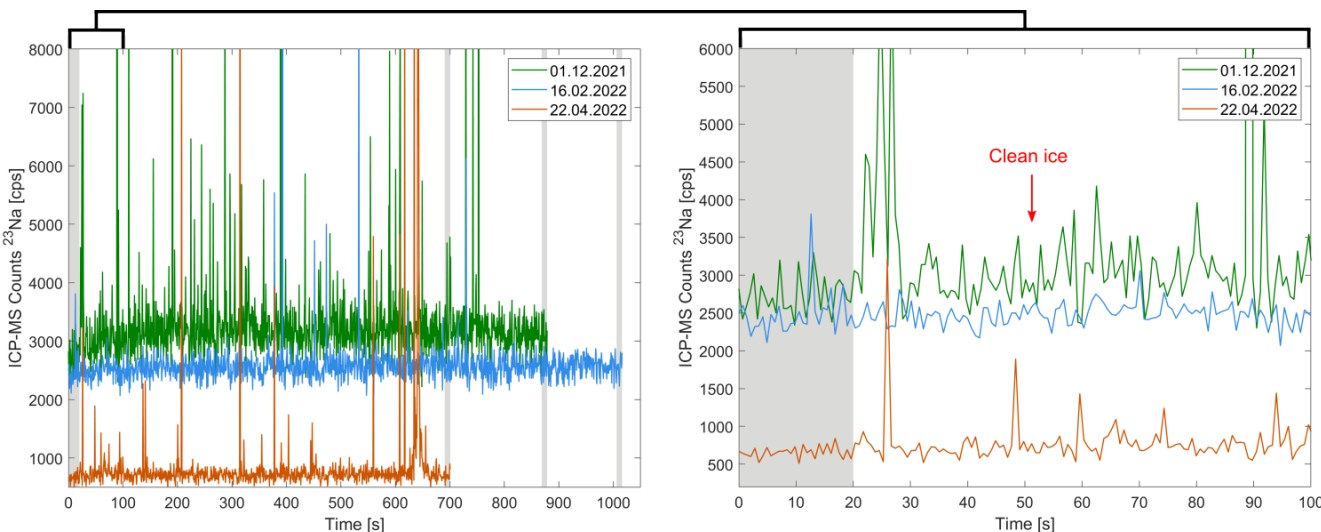

**Figure 5.** Sodium signals for three different blank ice samples made from MQ water. Left: Full paths (3-5 cm). Right: Zoom to the first 100 seconds of each measurement. The grey shading mark the recording of the system background with just the He-flow and no laser.

After the first 20 s, the laser is switched on, which typically results in an initial peak (see Fig. 5 right panel) caused by residual material in the sample lines or accidentally scanning across the outer edge of the sample. The average intensity of the signal retrieved from the blank ice is barely distinguishable from the one with just helium. Therefore no significant contamination could be detected. However, the MQ blank ice signal shows more variability and a larger amount of sharp high frequency peaks compared to the gas blank. The origin of these peaks is not entirely clear. The most likely source is microscopic particles that are either incorporated in the ice matrix or could not be sufficiently removed during the sample preparation process. The median of the Na signals from MQ ice is on average about 9% higher than the one of the helium stream. We conclude that this was low enough not to affect the measurements. For the other elements no systematic difference between only He and MQ ice




**Table 4.** Depth and age ranges of the available LA-ICP-MS data sets for Skytrain ice core. For each depth interval $^{23}$Na, $^{24}$Mg, $^{27}$Al and $^{43}$Ca were analysed together. The term yBP refers to years before 1950 AD (present). The age ranges and annual layer thickness (ALT) estimates were calculated using the ST22 chronology (Hoffmann et al., 2022; Mulvaney et al., 2022). For the last two depth intervals only approximate values are given, because an exact dating was not possible due to folding in the ice.

| Depth range [m] | Age range [yBP] | Depth resolution [μm] | Theoretical ALT [mm] |
|---|---|---|---|
| 83.2 - 84.0 | 558 - 565 | 185 | 110 |
| 355.2 - 355.6 | 6315 - 6332 | 185 | 24 |
| 395.2 - 395.6 | 8399 - 8430 | 185 | 13 |
| 468.0 - 468.8 | 20 448 - 20 823 | 185 | 2.1 |
| 480.8 - 481.6 | 26 179 - 26 768 | 82 | 1.4 |
| 609.3 - 609.6 | ≈ 112 000 | 82 | ≈ 1.5 |
| 613.6 - 616.8 | ≈ 115 000 - 117 000 | 82 | ≈ 1.5 |

could be detected. The MQ signals show characteristics of red noise, e. g. an exponentially decaying auto correlation function. This is typical for environmental background signals. Therefore, we conclude that the very high frequency signals and sharp peaks in Na intensity should be regarded as noise. These findings were taken into account in section 3.4 for interpretation of signals with respect to identification of layers. Very high peak values, at least ten times above the median were regarded as outliers and removed. As a general blank correction, the mean of the first 20 s pure helium signal was subtracted from the total signal for each measured element.

## 3.2 First results - comparison to CFA data

An overview of the chosen depth and age ranges of all analysed samples from Skytrain Ice core and the expected annual layer thicknesses based on Hoffmann et al. (2022) and Mulvaney et al. (2022) is given in Table 4.

In a first evaluation of the LA-ICP-MS data, they were qualitatively compared to CFA records on two different depth intervals, one shallow (∼ 560 - 570 yBP) and one deep (∼ 20.45 - 20.73 ka BP). This comparison aimed at the identification of common trends in both data sets and an evaluation for which elements the LA-ICP-MS technique produces reliable and significant signals.

A shallow section of 80 cm length between 83.2 and 84 m core depth consisting of 16 samples (5 cm each) was analysed. Based on the age model, in this depth and age range an annual layer thickness of about 11 cm is expected. A round laser spot of 150 μm diameter was used, which results in a depth resolution of about 185 μm. The results of these measurements are shown in Fig. 6. The coloured signals show the LA-ICP-MS results and the grey signal on top shows the CFA ICP-MS data (Grieman et al., 2021) for comparison. The laser data are reported in counts per second and not converted into calibrated concentration,




so only relative changes can be compared. They were corrected for the blank and outliers according to the procedure in section 3.1. For better readability the laser ablation data were smoothed with a 4.5 mm running average.

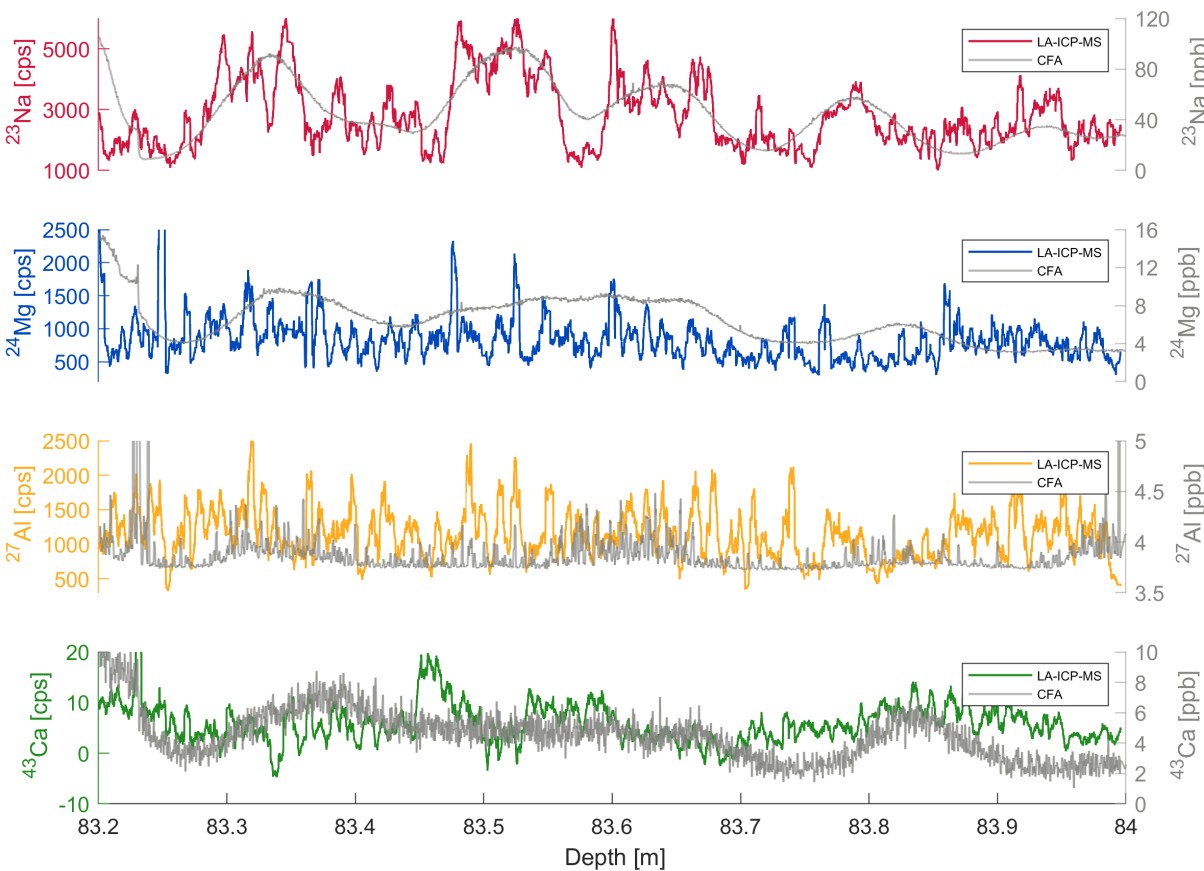

**Figure 6.** LA-ICP-MS signals of Na, Mg, Al and Ca from a shallow section (ca. 560-570 yBP) of Skytrain ice core. The smoothed high resolution LA-ICP-MS data (colored) is compared to the CFA ICP-MS data (grey). Note the generally good correlations for Na, Mg and Ca.

The LA-ICP-MS sodium data clearly follow the course of the low resolution CFA data, while even in the smoothed version they show much smaller scale (high frequency) variation than the the CFA signal. This finding indicates, that even at this shallow core depth the sodium CFA signal might not be able to resolve seasonal variations present in the ice. For magnesium and calcium the correlation is less obvious, but still visible. In particular the decrease in concentration between 83.2 and 83.3 m is well represented in the laser data. Also the large enhanced plateau between 83.4 and 83.65 m as well as the dip in concentration around 83.7 m can be identified. For aluminium the relation is less obvious. The CFA aluminium concentrations are close to the detection limit in this segment and some of variability may not be resolved. A direct comparison of these two



data sets therefore is not reasonable. These results demonstrate the capability of the LA-ICP-MS measurements to replicate larger trends and longer periods in concentration changes, but also to reveal much smaller scale variations, which cannot be resolved by conventional CFA methods. This is surprising because, according to the spatial resolution of $\sim 4$ cm of the CFA ICP-MS, it should in principle be possible to identify features smaller than the roughly 10 cm variations that are visible in the Na data. We attribute this remarkable smoothing of the CFA data mainly to turbulent mixing effects in the sample lines and the ion source of the CFA ICP-MS. To what extent the small scale variations in the laser ablation data can be diagnosed as (annual) layers will be discussed in sections 3.3 and 3.4.

To evaluate if the overall good correlation between LA-ICP-MS and CFA data is preserved in the deeper and older parts of the Skytrain ice core, a section from the Last Glacial Maximum (LGM) about 20 ka BP was analysed. The results for the LA-ICP-MS measurements compared to the CFA ICP-MS measurements are shown in Fig. 7. The theoretical expected annual layer thickness at this depth is about 2.1 mm and therefore much smaller than the resolution of the CFA data. Again, to increase readability the LA data were smoothed using a running average of 4.5 mm.

As in the shallow section, for Na an impressive coherence of the large scale variations of the LA-ICP-MS and CFA signals is visible. Also the general trend of the LA-ICP-MS Al signal follows the CFA. The two peaks at $\sim 468.45$ m and 468.6 m are especially well represented. The magnesium signal generally shows little variation in both data sets, which makes a direct comparison difficult. The LA-ICP-MS calcium signal suffered from a contamination issue in the ICP-MS at the time of these measurements. It is therefore not shown or discussed here. The CFA-signal shows much more high frequency noise in this section than in the shallow one, which is likely due to higher dust concentrations in the LGM. This leads to a greater quantity of individual dust particles being partly ionised in the plasma, generating sharp concentration peaks in the data. Superimposed on this increased noise level, the LA-ICP-MS signal exhibits more distinct small scale features. The meaning of these features need to be evaluated before interpretation in term of layers and climate signals (see sections 3.3 and 3.4). Overall, the findings from these comparisons between LA-ICP-MS and CFA-ICP-MS confirm the initial assumptions that (i) there is a common signal in the LA-ICP-MS and CFA ICP-MS data and (ii) the LA-ICP-MS data contain additional information on high frequency impurity variations in the shallow and deep core sections of Skytrain ice core.

## 3.3 Layer identification

### 3.3.1 Spectral Analysis

Based on the data comparisons between CFA- and LA-ICP-MS, the main challenge is to reduce the high frequency noise of the LA-ICP-MS data to extract meaningful information on periodic impurity concentration changes on different time scales and and assess their interpretation as stratification or layering. In the following, we focus on the sodium signal because it has the highest level above background and variations in amplitude.

To extract the additional information that is contained in the LA-ICP-MS signal compared to the CFA signal, we performed spectral analysis of selected depth sections. We used Fourier analysis similar to the approach in Bigler et al. (2011) and Wavelet analysis based on the method of Torrence and Compo (1998). First, we applied these methods to the shallow core section, to



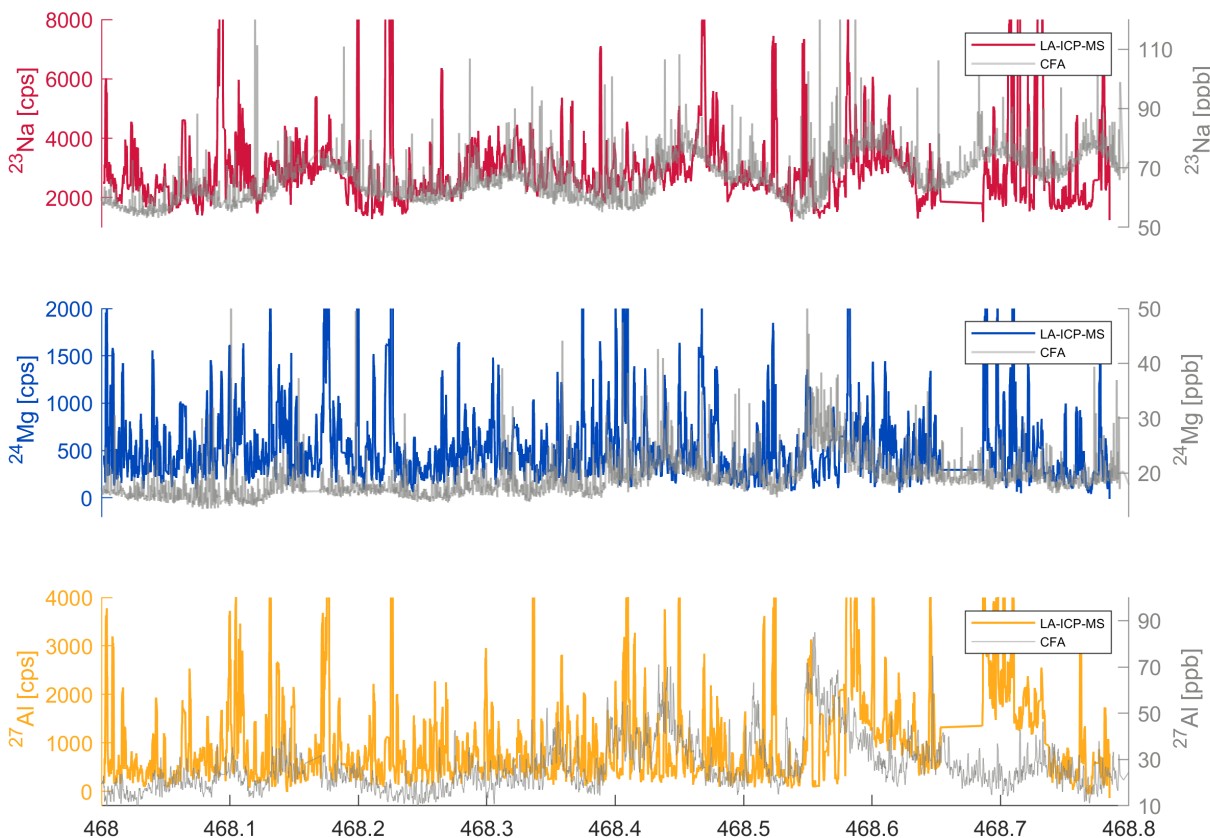

**Figure 7.** LA-ICP-MS signals of Na, Mg and Al from a deep section (ca. 20 450 - 20 730 yBP) of Skytrain ice core. The smoothed high resolution LA-ICP-MS data (colored) is compared to the CFA ICP-MS data (grey).

test if the results are in line with the expected layer thicknesses based on annual layer counting and the age model. We found
a striking alignment between the fluorescence calcium CFA data (depth resolution ∼ 1.4 cm) and the sodium LA-ICP-MS signal (see Fig. 8 left). Sodium is well known to show seasonal variations in Antarctic ice cores with peak concentrations in winter (Sigl et al., 2016). The calcium concentrations have also been found to show seasonality, however these signals were less distinct than sodium (e.g. Curran et al. (1998); Haines et al. (2016)).

To quantify the common frequencies of both signals, we performed Fourier transforms on the LA-ICP-MS Na and the CFA Ca
data. Butterworth low pass filters were applied to cut off frequencies above the respective depth resolutions of the instruments. The retrieved power spectra are shown in Fig. 8 in the right panel. Additionally, the single maximum of a sine function with the period length of the theoretical annual layer thickness (11 cm) from the age model (Hoffmann et al., 2022) is shown. There



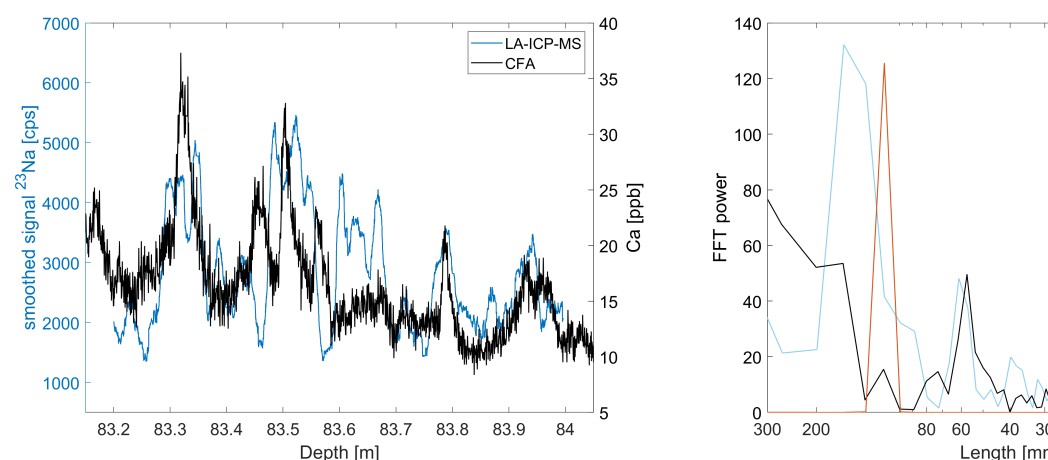

**Figure 8.** Left: Sodium LA-ICP-MS signal of a shallow core section of Skytrain ice core compared to CFA Fluorescence Calcium measurements from the same depth section. Note the striking alignment of both datasets. The sodium data was smoothed with a 1.4 cm moving average to mimic the resolution of the calcium CFA data. Right: Power spectrum of the fluorescence calcium data, the LA-ICP-MS sodium data and a sine function with the period length of the expected annual layer thickness in this core depth.

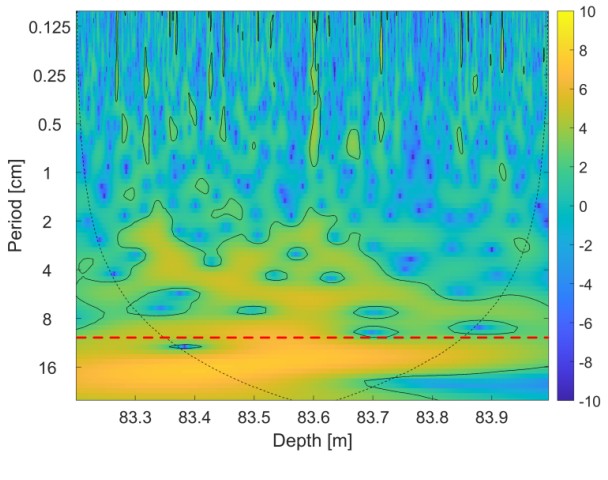

(a) LA-ICP-MS sodium wavelet spectrum

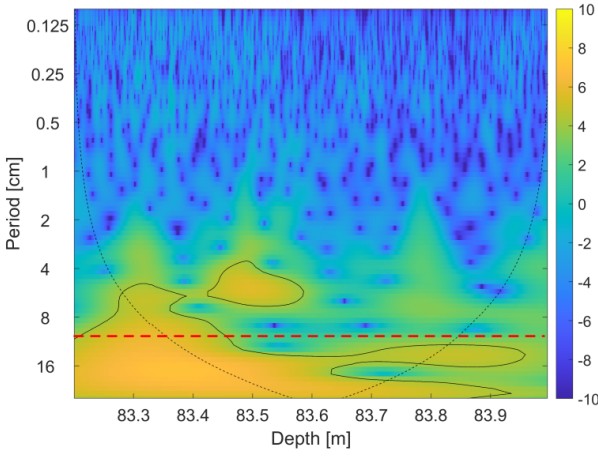

(b) CFA Fluorescence calcium wavelet spectrum

**Figure 9.** Wavelet spectra of unsmoothed LA-ICP-MS sodium and CFA Fluorescence calcium for a shallow section of Skytrain ice core. The solid black line denotes the 95% confidence level. The thin dashed black line marks the cone of influence (COI), data below that curve might be influenced The horizontal dashed line marks the annual layer thickness from the age model.

is a clear common maximum of Na and Ca around 6 cm and another one around 16 cm. This finding is also supported by the results of the wavelet analysis (see Fig. 9). Wavelet analysis was performed using a Morlet wavelet and an assumed red-noise



background with a lag-1 autocorrelation coefficient of 0.22 for the LA-ICP-MS data. This resulted from the autocorrelation
analysis of the MQ blank samples (see section 3.1). For the CFA data, a standard coefficient of 0.72 was used. The solid black
lines in Fig. 9 denote the 95% confidence level based on the red noise background correction. The thin dashed black line marks
the cone of influence (COI). Data below the COI might be influenced by boundary effects. Similar to the fourier spectra, there
are two areas of enhanced power in both data sets, one between about 4 cm and 8 cm and the other around 15 - 16 cm period
length. The age model in this core section is based on annual layer counting of the CFA Na and Ca signals constrained by
absolute age markers, mainly volcanic eruptions at this depth. The second maximum at ∼ 16 cm is also the most dominant
period in both wavelet spectra (see Fig. 9). The results of the power spectra therefore disagree with the expected modeled layer
thickness of 11 cm. This could hint that the age model might slightly either overestimates or more likely underestimate the
annual layer thickness in this particular 80 cm section of ice. The results of the spectral analysis of the LA-ICP-MS sodium data
also corroborate the good agreement with the CFA data. They additionally show the potential to reveal smaller scale variations,
e.g. an intermittent power enhancement in the sodium wavelet spectrum around ∼ 0.5 cm period length, which cannot be
resolved by the CFA even in this shallow core section.

### 3.3.2 Signal Stacking

The main challenge of identifying features in the LA-ICP-MS data, which can be interpreted as layers, is to separate the actual
signal from the noisy background. This is additionally complicated by the fact that some impurity species, especially sodium,
concentrate in the crystal grain boundaries e.g. (Barnes and Wolff, 2004; Eichler et al., 2017; Stoll et al., 2023). When the laser
ablation path crosses a grain boundary, the enhanced concentrations can cause sharp peaks in the LA-ICP-MS data which then
lead to misinterpretation of grain boundaries as larger scale features.

To solve this problem and make the identification of periodic layering possible, we made the following assumption: if layers
are horizontally preserved in the ice core, parallel laser paths along the depth axis of a sample should exhibit common intensity
features. To test this hypothesis, we performed detailed measurements on a 5.5 cm core section from 355.775 - 355.82 m depth.
From the age model, an annual layer thickness of about 2.4 cm can be expected at this depth.

We measured 8 parallel lines with 1 mm distance between each on the same sample. Pictures of the sample before and after
measurement can be found in Fig. A1. The results are shown in Fig. 10 (left). It is obvious that there is a common peak in
intensity in all lines between about 355.78 and 355.795 m, which strongly hints towards a layering feature. To better assess
this visual identification, we normalised and stacked the signals of all eight lines (Fig. 10 right). We also applied a low pass
filter to cut off the high frequency instrumental noise above the theoretical depth resolution of 185 μm (dark blue line). The
red line denotes a sine function with a period length of 2.4 cm, representing the theoretical annual layer thickness derived from
the age model. The main trend of the LA-ICP-MS signal clearly follows the sine function, especially between 355.775 m and
355.800 m. At 355.812 m there was a break in the sample which might lead to some boundary effects. However, even the
filtered signal shows superimposed higher frequency variations. These small scale variations can be caused by (i) incomplete
decontamination of the sample surface, (ii) particles that show up as sharp peaks in the ICP-MS signal (iii) actual small scale
intensity variations e.g. caused by accumulation of Na in crystal grain boundaries. Based solely on the line data presented



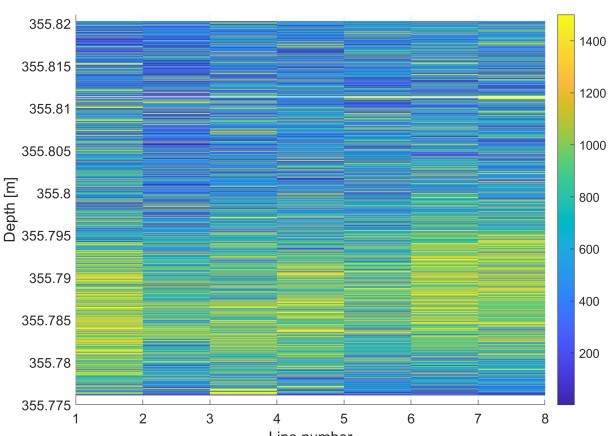
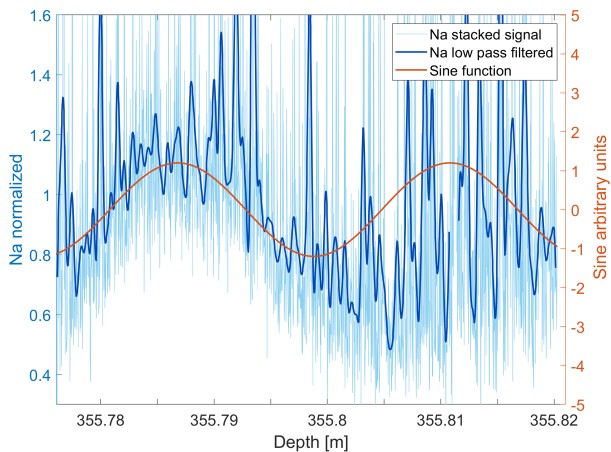

**Figure 10.** Left: Sodium signal of 8 parallel laser lines on the same sample (355.775 - 355.82 m), separated by 1 mm each. Common variations in signal intensity are clearly visible. Right: Stacked and normalised sodium signal of all eight lines. The dark blue line denotes a low pass filter cutting off the high frequency noise above the technical resolution of the instrument. The red line is a sine function with a period length of 2.4 cm representing the theoretical annual layer thickness. A clear correlation between the variation of the filtered data and the sine is visible.

here, which of these possible sources is the most dominant cannot be determined. However, the strong alignment of the main

LA-ICP-MS data trend with the theoretical layering shows that features of similar wavelength to the estimated annual layer thickness are still present at this depth of the core (about 6.5 ka BP) and that this layering can be revealed by the laser ablation technique.

### 3.4   Signal interpretation in deep ice

In the deepest parts of the ice core the modeled annual layer thickness is in the millimetre range or even below. To investigate if

it is still possible to identify layered structures in these depths, provided that they are preserved, we applied the spectral analysis methods discussed in section 3.3. Here we present two examples. The first is from a depth of 481.0 - 481.8 m representing about 26.8 ka BP. The resolution of the laser ablation measurements at this depth was 82 μm. Additionally, within this depth interval in a 3.5 cm section (481.101 m - 481.136 m) three parallel lines at 1 mm distance were measured along the depth axis of one ice core sample. One centimeter excerpts of the normalised and stacked Na, Mg and Al signals of these 3 parallel lines

along with a sine function representing the modeled layer thickness are shown in Fig. 11 and B1.

Wavelet analysis using the parameters determined in section 3.3 was performed on an single laser ablation line in the depth range of 481.10 - 481.35 m as well as on the stacked signal from the 3.5 cm section within this depth range. The results of this analysis for sodium are shown in Fig.12 on the left. The spectrum of the long single laser ablation line shows three areas of enhanced power, one between about 2 and 4 mm period length, one between about 8 and 12 mm and one around 32



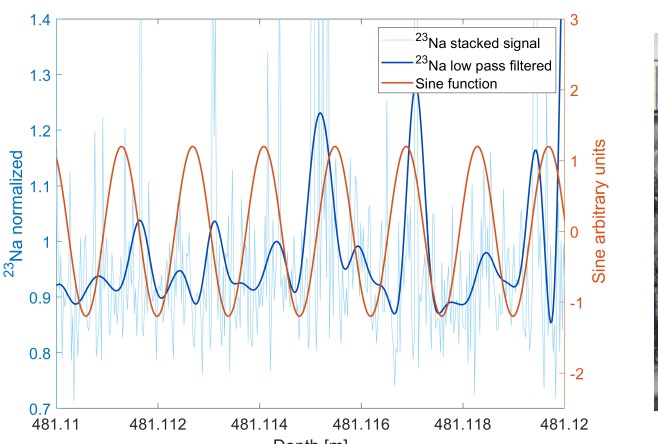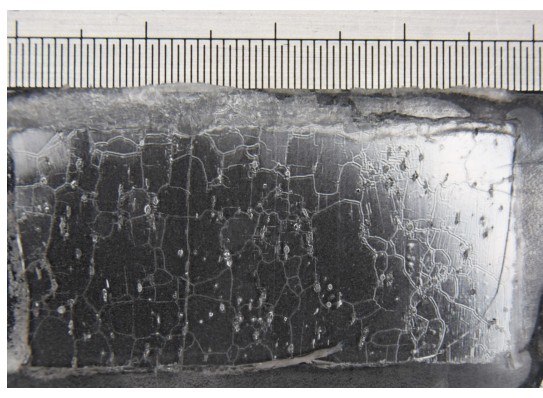

**Figure 11.** Left: Normalised and stacked sodium LA-ICP-MS signal of three parallel laser lines for 1 cm of Skytrain ice core about 26.8 ka BP. The dark blue line denotes a low pass filtered version of the data and the red is a sine function with 1.4 mm period length representing the theoretical annual layer thickness. Right: Sample picture of a 3.5 cm section 481.101 m - 481.136 m. The crystal grain boundaries are clearly visible. The scale is in cm.

mm period length. These areas were determined by counting and summarizing all power values above the 95% significance level for each period and calculating the area fraction covered by these data points over the whole sample depth (see Fig. 12 right bottom panel). Comparing the 2-4 mm periodicity to the sample picture in Fig. 11, suggests that this periodicity is likely to correlate with the estimated average crystal grain size in this depth. The 2-4 mm periodicity therefore probably does not represent layering in the sense of larger scale horizontal features, but accumulation of sodium in the crystal grain boundaries. The spectrum of the short stacked signal (Fig. 12 top left) shows a significant enhancement around a period length of about 1 mm, but only very intermittent. This could be interpreted as a hint at local preservation of periodic concentration changes superimposed onto the grain boundary effects in the sodium signal in the range of the expected annual layer thickness. This would require a significant amount of Na not being accumulated in the grain boundaries. However, this interpretation has to remain speculative at the present stage of investigation. Measurements of a larger number of parallel lines (see section 3.3.2) spaced wider than the average grain size would be needed to corroborate these observations and are planned for future analyses. Additionally to the wavelet analysis, Fourier analysis was performed on the stacked raw data, the low pass filtered data and for comparison also on the artificial sine function (Fig. 12 top right). The fourier spectrum shows an area of enhanced power at around 0.5 - 1 mm period length. This could support the findings from the wavelet analysis of the stacked signal.

The wavelet results for magnesium (Fig. 13) show approximately four areas of enhanced power, at about 1-1.5 mm, 3-6 mm, 8-12 mm and around 32 mm (see also Fig. C2a). The fourier spectrum (Fig. 13 right) of the stacked short lines generally supports this finding, with additional areas of enhanced power at about 0.20 - 0.275 mm and 0.6 - 0.8 mm. Magnesium is known to show a similar seasonality as sodium (e.g. Curran et al. (1998)). During the glacial, the total magnesium content contains a large fraction originating from mineral dust particles (de Angelis et al., 2013), which is not the case for sodium. The maxima in the



(a) Sodium Wavelet spectrum stacked lines

(b) Sodium Fourier spectrum stacked lines

(c) Sodium Wavelet spectrum single line

(d) Fraction of significant area

**Figure 12.** Left panels: Sodium Wavelet spectra of a single and three stacked LA-ICP-MS lines across a 25 cm section between 481.10 and 481.35 m. The dashed line marks the modeled annual layer thickness and the shaded areas the sections of enhanced power. Right top: Fourier spectrum of sodium from the stacked laser ablation lines. The dark blue is the raw stacked signal and the red the frequency peak of a sine function with the wavelength of the expected annual layer thickness in this section. An area of enhanced power between about 0.6 and 1 mm period length is visible (shaded area). Right bottom: Fraction of wavelet spectrum above the 95% significance level for each period length summarized over all depths. Three sections of enhanced power between ∼ 2-4 mm, 8-12 mm and ∼ 32 mm are standing out (shaded areas).







(a) Magnesium Wavelet spectrum

(b) Magnesium Fourier spectrum

(c) Aluminium Wavelet spectrum

(d) Aluminium Fourier spectrum

**Figure 13.** Left panels: Wavelet spectra of a single LA-ICP-MS line across a 25 cm section between 481.10 and 481.35 m for Magnesium and Aluminum. The dashed line marks the modeled annual layer thickness and the shaded areas enhanced power. Right panels: Fourier spectra of magnesium and aluminium from 3 stacked laser ablation lines in a depth of 481.101 m - 481.136 m. Green and black are the raw stacked signal and red a sine function with the wavelength of the expected annual layer thickness. The shading highlights sections of enhanced power.

fourier spectra at very small wavelengths of about 250 µm could hint to effects generated by ablation of such single particles. The soluble magnesium fraction can migrate into crystal grain boundaries (e.g. (Bohleber et al., 2021)). We therefore attribute the larger periods (3-6 mm and 8-12 mm) again to grain boundary effects. It is remarkable, that the magnesium data shows





an additional area of enhanced, although as well strongly intermittent power, around $\sim 1.5$ mm period length and thus in the range of the annual layer thickness from the age model. In summary, these findings could be an indication that the magnesium is less affected by grain boundary effects than the sodium. In consequence, magnesium might be better suited for identification
of annual layers, but this needs to be confirmed by more detailed analysis (parallel lines and signal stacking) in the future.

The aluminium wavelet spectral data (Fig. 13) show less isolated areas of enhanced power compared to sodium and magnesium. There is a zone of higher power around 8-16 mm, and also around 30 mm (Fig. C2a) but for other wavelengths the identification of more enhanced power sections becomes speculative. Also the fourier spectrum (Fig. 13) shows less outstanding sections compared to Na and Mg. There seem to be some enhancements around 0.25 mm, 1.3 mm and 4 mm. Aluminium is mainly of
terrestrial dust origin and the LA-ICP-MS signal is supposed to be much more influenced by discrete particles than Na and Mg. Therefore the LA-ICP-MS data are expected to exhibit more sharp excursions and peaks and are potentially less influenced by grain boundary aggregation. The maxima in the fourier spectra at very small wavelengths of about 250 μm could again hint to such effects. Overall, there are two important findings resulting from these spectral analyses: First, the sodium and magnesium signal at an age of $\sim 27$ ka BP in Skytrain ice core are likely influenced and probably dominated by grain boundary effects.
Second, there are indications that consistent periodic concentration changes in the range of the expected annual layer thickness, superimposed to the grain boundary effects, might be preserved and can be revealed by the laser ablation technique.

The second example that will be discussed regarding signal interpretation in deep ice is a 25 cm section from the oldest part (LIG) of Skytrain ice core (615.35 m - 615.6 m). The modeled layer thickness in this section is expected to be $\sim 1.5$ mm. The spatial resolution of the laser measurement was 82 μm. It should be mentioned that between about 605 m and 617 m some
disturbances occur in the ice structure, probably due to some larger scale folding and repetition of layers (Mulvaney et al., 2022). Therefore, this section serves as an example of a challenging setting for data interpretation.

For this section, only single laser ablation lines on each 5 cm sample were measured, so no stacking of data were possible. The LA-ICP-MS sodium data is presented in Fig. B2. The blue signal is the LA-ICP-MS sodium and the red signal is the corresponding CFA sodium data. Both data sets follow the same large scale trend. The wavelet power spectrum of the LA-
ICP-MS sodium data is shown in Fig. 14 on the left. Several zones of enhanced power can be identified, however none of them continuous over the whole depth interval: $\sim 1.5$ mm period length, 6 mm, 12 - 20 mm and about 24 mm (Fig. C1b). The average estimated crystal grain size (see sample pictures in Fig. A2) in this depth section is on the order of about 3 - 6 mm. There are some grain sizes up to 10 mm, grains smaller than 2 mm are sparse. We therefore attribute the enhanced power in the periods larger than 2 mm to grain boundary effects. The power maxima around 1.5 mm wavelength, the one at $\sim 0.5$ mm and others
below however cannot be explained by these effects. The Fourier spectrum (Fig. 14 right) as well as the wavelet spectrum shows a rather distinct peak at 1.5 mm wavelength. The largest peak however is at a period length of about 6 mm and an additional one at about 12 mm. This indicates that despite large influences of the sodium accumulation in grain boundaries, there might be some underlying features that are unaffected by this phenomenon. However, there is no continuous power enhancement across the whole depth interval and thus if these features can be interpreted as layering it is only preserved very locally. Especially in
the depth section between 615.40 - 615.45 m significant frequencies are sparse. Therefore, the enhanced spectral power in the range of the modeled annual layer thickness could again only hint towards a possibly partial preservation of layering at this





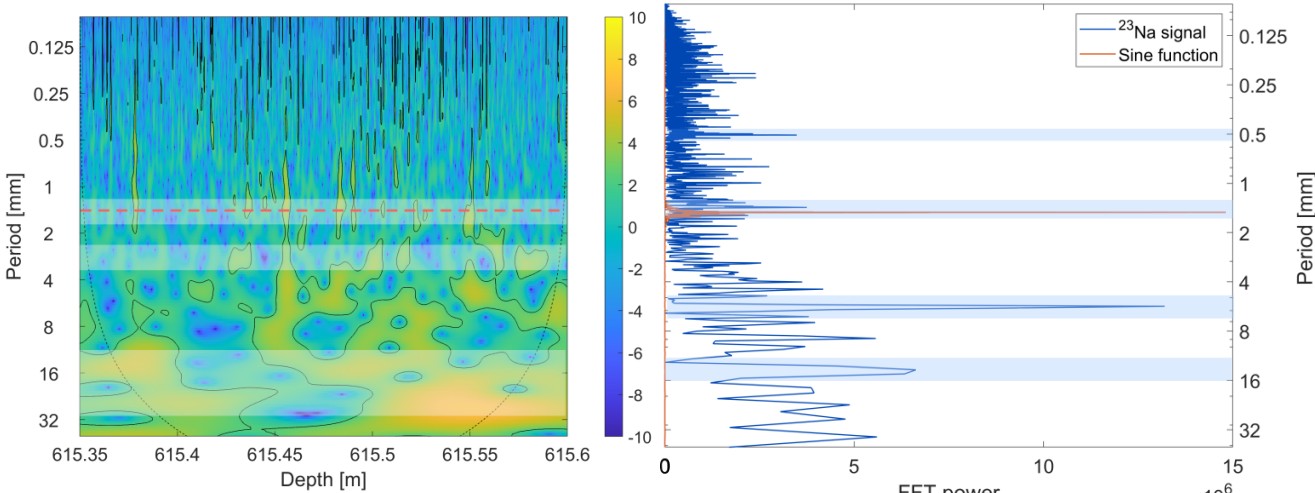

**Figure 14.** Left: Wavelet spectrum of sodium of a single LA-ICP-MS line across a 25 cm section between 615.35 - 615.60 m. The dashed line marks the modeled annual layer thickness and the shaded areas enhanced power. Right: Fourier spectrum of sodium (dark blue) for a single laser ablation line in a depth of 4615.35 - 615.60 m. The red is the frequency peak of a sine function with the wavelength of the expected annual layer thickness the shading highlights sections of enhanced power.

depth of Skytrain ice core. To further corroborate this finding, it would be necessary to ablate parallel lines and use the stacking technique discussed above to reduce the noise and grain boundary effects, which will be done in future analyses. However, this initial analysis looks promising and shows that the LA-ICP-MS measurements are capable of revealing potentially layered structures in depths of Skytrain ice core where conventional methods fail. Nevertheless, due to the possible disturbance of the ice structure in this depth region, a climatological interpretation based on these signals alone is not unambiguously possible.

## 4 Conclusions

A system for LA-ICP-MS measurements of glacier ice samples was successfully installed at the Department of Earth Sciences of Cambridge University. The system can reliably operate with a spatial resolution of up to 80 μm on ice samples. In the current setup, $^{23}$Na, $^{24}$Mg, $^{27}$Al and $^{43}$Ca were analysed simultaneously. The signal strength of $^{23}$Na, $^{24}$Mg and $^{27}$Al proved to be well above the instrumental background for most samples. The calcium signal however was lost in the noise for most measurements. This problem can potentially be solved by use of helium in the collision cell of the ICP-MS to lower the background signal, which will be tested in future applications. The LA-ICP-MS method was used to analyse Skytrain ice core from West Antarctica. It was found that the longer wavelengths of the LA-ICP-MS signals correlate well with the lower resolution CFA data. This proves that the method is capable of producing real, stable, reproducible signals. Spectral analysis of the high resolution LA-ICP-MS sodium data revealed that although the dominating frequencies are probably influenced by crystal grain boundary effects, there are significant shorter wavelengths that hint towards a preservation of layered structures superimposed onto the





effects of impurity accumulation. Thus, there are indications that it could be possible to identify annual layering in the ice core sodium and magnesium signal even in deep ice. This is an important finding also for other projects like "Beyond EPICA",

where a reliable analysis of the deep ice structures is crucial for meaningful climate related interpretation. Nevertheless, the frequency analysis cannot conclusively and unrelated to other information be used to identify horizontal layering. Therefore interpretation of the high frequency LA-ICP-MS signal components remains challenging. Additional information about the expected layer thickness, the average crystal grain size and the composition of the measured species regarding particulate and soluble fractions is needed to interpret the retrieved power spectra. It was found that signal stacking of parallel laser lines

on the same sample is a powerful tool to reduce noise and better extract meaningful variations in the frequency analysis. It is therefore highly recommended that at least two or more parallel measurements on each ice sample, spaced wider than the average grain size are performed in the future. Extensive measurements of parallel lines on one 5 cm sample could hereby also help to interpret the signals of single lines from adjacent depth intervals. We conclude that layered structures in the period length of the expected annual layer thickness in sodium and magnesium are potentially preserved in Skytrain ice core to an age

of at least 26 ka BP and probably also the LIG. These findings have the potential to open a new field of possibilities for climate data interpretation with respect to fast changes based on high resolution LA-ICP-MS data in the future.



## Appendix A:  Sample pictures

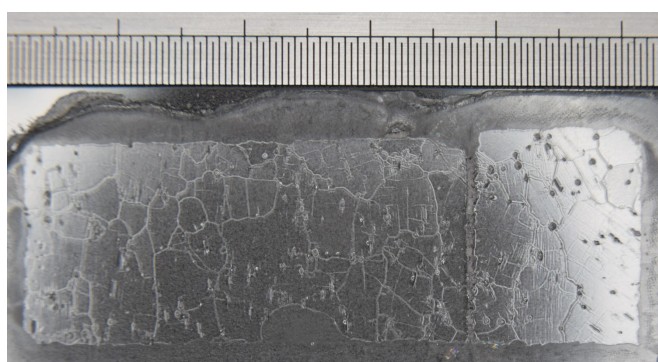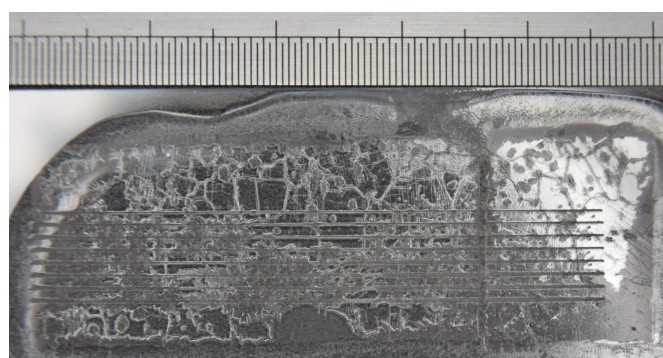

**Figure A1.** Left: Picture of a section of Skytrain Ice core from 355.775-355.82 depth previous to LA analysis. The scale is in cm. Right: The same section as on the left after LA analysis. The ablation paths of 8 parallel lines are visible. The blurred lines are a result of sublimation during and mostly warming after analysis.

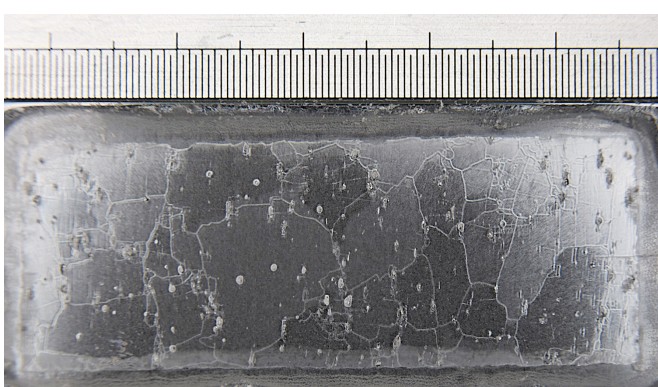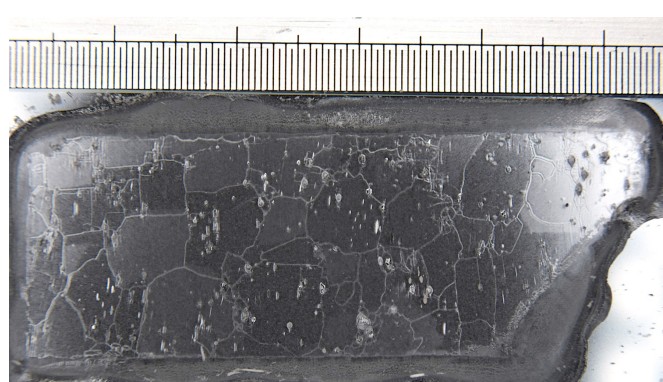

**Figure A2.** Pictures of two 5-cm-samples from Skytrain ice core from a depth of 615.20 - 615.25 m (left) and 615.40 - 615.45 m (right). The scale is in cm. Note the large ice crystal grain sizes in the order of ∼ 3-10 mm.



## Appendix B: LA-ICP-MS data

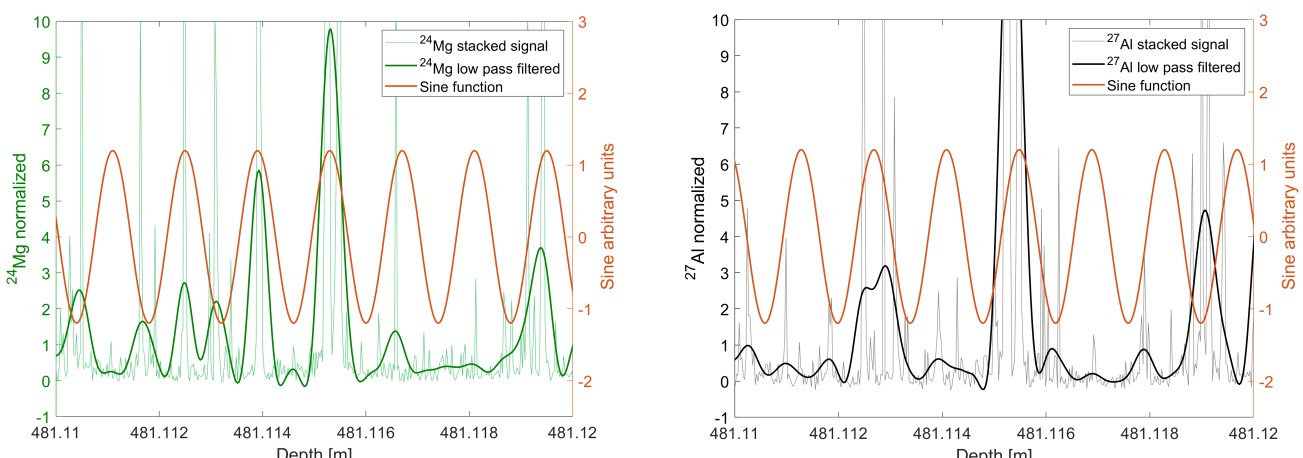

**Figure B1.** Left: Normalised and stacked magnesium LA-ICP-MS signal of three parallel laser lines for 1 cm of Skytrain ice core about 26.8 ka BP. Sine function of 1.4 mm wavelength for comparison. Right: Aluminium data for the same depth section.

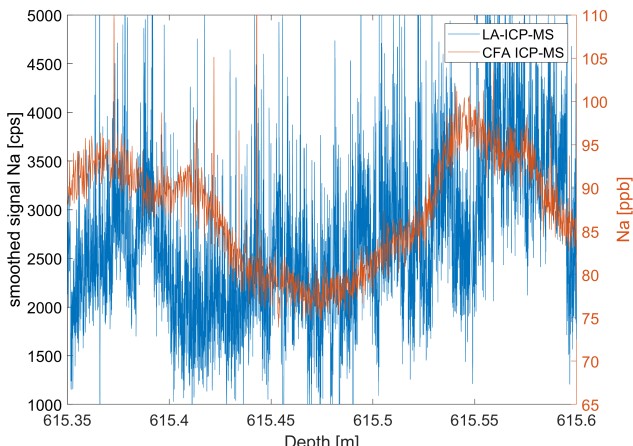

**Figure B2.** Sodium LA-ICP-MS data from a section between 615.35 - 615.60 m of Skytrain ice core. Blue is the LA-ICP-MS sodium data smoothed to the laser resolution, red the corresponding CFA sodium data for this section. Note the common trend of both data sets.



## Appendix C: Wavelet analysis - significant sections

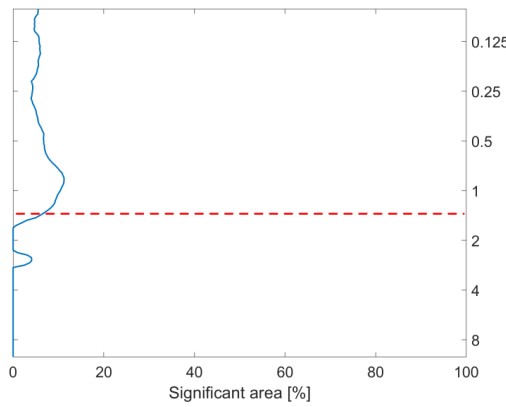

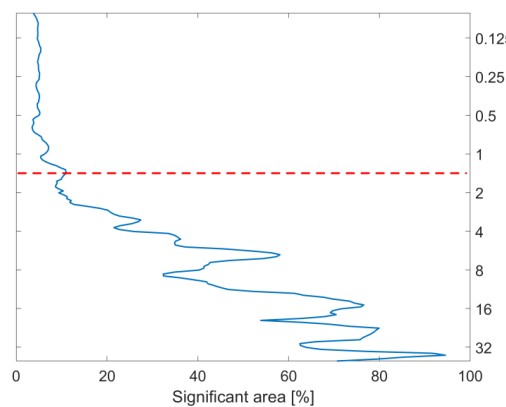

(a) Depth: 481.101 m - 481.136 m. The highest values are located in the vicinity of the modeled annual layer thickness.

(b) Depth: 615.35 - 615.60 m There is a slight enhancement in the vicinity of the modeled annual layer thickness.

**Figure C1.** Fraction of wavelet spectrum of the sodium signal from two different depth intervals above the 95% significance level for each period length summarized over all depths.

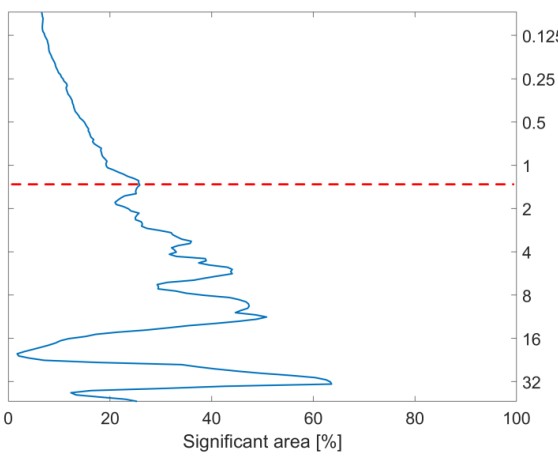

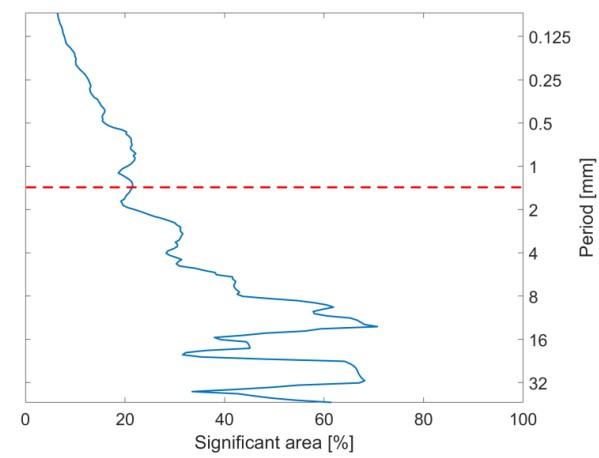

(a) Magnesium data. Four sections of enhanced power between $\sim$ 1-1.5 mm, 3-6 mm, 8-12 mm and around $\sim$ 32 mm are visible.

(b) Aluminium data. There are two enhanced power sections at about 8-16 mm and 30 mm.

**Figure C2.** Fraction of wavelet spectrum of magnesium and aluminium from a single laser ablation line in a depth of 481.101 m - 481.136 m above the 95% significance level for each period length summarized over all depths.



*Author contributions.* The paper was written by HH, with contributions mainly by RR, EW and MG. The ice core was drilled and sectioned by EW, RM, CNA, MMG and IR. The CFA analysis was performed by HH, MG, JH, RM, RR, and IR. The laser ablation measurements were performed by HH and JD. SG provided expertise on the laser ablation measurement technique. All authors contributed to improving the final paper.

*Competing interests.* No competing interests are present

*Acknowledgements.* The authors thank Liz Thomas for access to the BAS ice lab facilities and general support of the project. The authors also thank Shaun Miller, Emily Ludlow, and Victoria Alcock for help with cutting and processing the ice core and Charlie Durman for ice core preparation for the CFA analysis. This project has received funding from the European Research Council under the Horizon 2020 research and innovation programme (grant agreement No 742224, WACSWAIN). This material reflects only the author's views and the Commission is not liable for any use that may be made of the information contained therein. EW and HH have also been funded for part of this work

through a Royal Society Professorship.



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
