# Peer review of "Laser Ablation - ICP-MS measurements for high resolution chemical ice core analyses with a first application to an ice core from Skytrain Ice Rise (Antarctica)"

_EGUsphere, 2023_

## Author Comment (AC1)

**Replies to reviewer 1 comments in calibri italic**

*Thank you for your review and your valuable input. We agree that there were some technical inaccuracies and especially the signal interpretation section needed some clarification.*

General comments

Hoffmann et al. present a new laser ablation inductively-coupled plasma mass spectrometry (LA-ICP-MS) setup for ice core analysis, established at the University of Cambridge. They analyzed several meters of the Skytrain ice core from coastal Antarctica in order to demonstrate the capability of the setup to detect layers thinned beyond the resolution of continuous flow analysis.

*We note that although Skytrain Ice Rise is 'coastal' relative to the majority of Antarctic ice cores, it is actually ~700 km inland and as a result has low impurity concentrations (lower than EPICA Dome C for example). The analytical challenge regarding the limit of detection and background is substantial compared to true coastal locations.*

Setups for LA-ICP-MS ice core analysis have now been established by several groups following an early pioneering phase in the 2000s. The interpretation of the high-resolution LA-ICP-MS ice core signals is not straight-forward but by now the technique has demonstrated to offer new means to investigate both, highly thinned stratigraphic signals and the ice-impurity interaction at the grain scale. However, compared to meltwater analyses only limited data exists so far and clearly there is still much to be learned from ice core analysis by LA-ICP-MS. It is thus of interest to the ice core community to see another group invest in this technique, in the hope that additional data obtained with different setups will also provide additional insight for various ice conditions.

To fully demonstrate that the new setup can meet this expectation the work presented here could, and should, be much improved. In its present state the manuscript is not able to fully capitalize on the data, since substantial ambiguities remain about the production of the data and the interpretation. Considering the amount of work that very obviously has gone into the measurements, this is regrettable and hopefully can be remedied. In the following I am trying to provide concrete suggestions on where further work is needed in order to strengthen the manuscript. I will primarily focus on two main areas and neglect other smaller issues at this stage.

The first main area concerns the LA-ICP-MS setup and the analytical details regarding the measurement of ice core samples. Since this is the first paper about this system the analytical components matter and need to be well documented. The following points need to be addressed and clarified:

A fundamental issue seems to be that, based on the statement made in line 160, the laser did not couple reliably to the ice and that settings had to be constantly adjusted in order to achieve ablation. Is this why not one set of parameters by many were used for analysis? One would expect just one settings of ablation parameters being used (apart from preablation) like in previous studies targeting line profiles (e.g. Della Lunga et al., 2017).

*The laser did couple reliably with the ice. Only for the completely bubble free, totally transparent blank ice samples the laser settings needed to be adjusted, for example lower scanning speed and larger spot size was required. The settings for the glacier ice sample*

*measurements were optimised at the beginning of every measurement session, which usually lasted 3-4 days with several weeks to months in between. During this time, the laser ablation system was not only used for ice but for geological samples and was subject to the normal processes of usage and wear. This led to the need of readjustment of the major settings (laser power, repetition rate) for each measurement session. Within the session, the settings were kept constant, to ensure that each continuous set of 8 or 16 consecutive ice core samples were analysed with the same parameters. We amended section 2.3.3 accordingly, (L 162 ff revised manuscript).*

In the manuscript the reported fluences vary strikingly between spot sizes: 6-8 J cm$^{-2}$ (Table 2 states 6 J cm$^{-2}$) for the "round" spot versus 36-55 J cm$^{-2}$ (Table 2 states 36-50 J cm$^{-2}$) for the rectangular spot. Is the laser really capable of providing 36-55 J cm$^{-2}$ or is this a mistake? How could such a large difference in fluence needed for ablation be caused by the choice of the spot geometry, isn't the beam homogenized and filtered by a mask to generate various spot sizes and geometries?

*The fluence was 6-8 J cm$^{-2}$ for all spot geometries and sizes with only minor variations. The high numbers for the rectangular spot geometries were incorrect and based on an erroneous readout by the laser software, which was recently confirmed by the manufacturer. We corrected the values in the manuscript accordingly.*

There are many instances throughout the manuscript where it is not clear for the reader what was done precisely and how. This needs to be made more traceable in order to understand which settings were actually used for data production.

*We corrected the values in Table 2 and also added the specific laser repetition rates, pre-ablation settings, spot sizes and fluences used for each depth interval in Table 4.*

Section 2.3 is an example in this regard, with predominant use of reporting ranges (e.g. 20 – 150 Hz) and "typical" settings ("*round spot of typically 150 µm*"). These statements do not match the values reported in Table 2, which adds to the confusion.

*There were some small errors in the reported ranges and numbers. We corrected them and reformulated parts of section 2.3.3 to describe the procedures more precisely.*

Considering that fluences of 6-8 J cm$^{-2}$ were needed for the "round" spot, this is significantly higher compared to previous LA-ICP-MS ice core analysis using a 193 nm excimer laser, where only 2-4 J cm$^{-2}$ were required (e.g. Müller et al., 2011). This is an interesting point that deserves more attention: Is the large fluence needed due to the setup, or due to the ice? What happens if comparable settings to previous studies are used: No ablation visible or some ablation but not enough signal in the ICP-MS?

*We agree that it is higher than the fluences used in other setups. We found that with the configuration we used (e. g. 40 µm/s scanning speed, 20Hz repetition rate, we needed a fluence above 6 J cm$^{-2}$ to make the ablation happen. This might be due to the finely microtomed and thus much smoother, more reflective and less uneven surface of the samples compared to other systems.*

A related important point concerns the complete absence of any optical close-up images taken with the laser on-board camera (which should exist in this setup). Considering the general importance of such images, it is hard to understand why they are not mentioned here.

*We agree that close up pictures can help to assess the surface conditions of the sample and we did provide them in the manuscript (appendix). The quality of the on-board camera pictures of the laser is not significantly better than the high resolution photographs of the samples before and after measurement. See comparison of large scale picture (2 x 5 cm) to example overview picture from Bohleber et al.( 2023) and on-board camera pictures below. We can share selected full resolution pictures of all samples in the supplementary information if needed and add an exemplary mosaic picture of the laser camera in the appendix (see below, 2 x 3.8 mm, vertical 150 μm ablation path on the left)*

[Figure]

[Figure]

*Overview pictures from different systems compared: A picture of a 2 x 5 cm sample from ~ 100 ka BP from Skytrain ice core on top, grain boundaries and bubbles clearly visible. Bottom overview picture from Bohleber et al. 2023. Note the difference in surface roughness.*

[Figure]

*Left: 2 x 3.8 mm mosaic picture of the on-board camera of the Cambridge laser system compared to an on-board camera picture from the Venice system (Bohleber et al. 2023) on the right.*

Optical close-up images would not only help significantly also for data interpretation (see below) but should also demonstrate how ablation craters look like, in particular when using such high fluences and large spots. In addition, the optical appearance of the surface would be important to show with regards to sample preparation, where interesting differences exist compared to previous studies.

*We added an exemplary close up image of the ablation path with the 150 μm round spot in Fig. 3 (see below). The surface in general appears to be smooth, with no visible contamination like for example scratch marks or particles.*

[Figure]

*Fig. 3 addition: close up image of laser path and grain boundary*

After sample preparation in the cold room, the samples seem to be actually stored for a significant amount of time until they are measured. There are some hints that contamination could not be fully avoided (see below).

*We cannot imagine how storage alone while the sample surfaces are not in contact with any material would lead to increasing contamination over time. The final preparation step with the sledge microtome creates a perfectly smooth surface, which is not touched by any material until the laser ablation measurement in the cryocell. The handling of the samples during insertion into the cryocell was done in a dry nitrogen purged glovebag to minimise any deposition of dust particles or condensation on the sample surface. We avoided another step of manual surface scraping before insertion into the cell, first because of the risk of damaging the perfectly smooth surface, which could then lead to focusing problems of the laser.*

*Second because the risk of introducing more contamination by another manipulation of the surface and additional potential frosting was considered much higher than the potential theoretical benefits. The only realistic contamination possible are loose dust particles settling on the sample surfaces after the final preparation step. These particles would be removed during pre-ablation at the latest.*

Moreover, it would be of interest how much the ice surface changes visually after the final preparation step until measurement, e.g. by sublimation and associated widening of the grain boundaries on the surface? This could be important to better assess the impact of such matrix-related features on the signals obtained. The overview photos do not provide enough detail for this purpose.

*The samples were stored no longer than three days at -25°C. Longer term storage was done at liquid-nitrogen temperature, minimising sublimation (see L 89 ff revised manuscript). During the short-term storage time, we observed only minimal widening of the grain boundaries by sublimation, but the effect was very small. We observed grain boundary widths of about 5µm at maximum (see Fig. 3).*

There are discrepancies in the presentation of the washout time. Regarding the wide range in ablation settings used it should be addressed what effect these parameters had on the washout time, e.g. depending on element, spot size, geometry, fluence, etc. The so-called single pulse response (SPR) is typically measured as FW0.1M or FW0.01M (full-width at 10% of the maximum, or 1%, respectively) of single laser shots. In Figure 4, all peaks from a single spot show a FW0.1M in the range of 300-400 ms or more. This is inconsistent with the values reported in Table 3. Why does the DCI change the peak shape into a slow uptake (it seems $^{238}$U is almost going into saturation?). Further attention should be devoted to the fact that the washout times differ greatly between $^{238}$U and $^{23}$Na – what could be causing this, are the washout times different for all elements? It is stated in line 148 that the washout time was determined with a 50 µm spot – the rectangular one with fluence >36 J cm$^{-2}$? What fluence was actually used for the SPR experiments on NIST glass, the same as for ice?

*There were some errors in the presentation of the washout experiment. We recently repeated the experiment with the DCI, replotted the data, recalculated the washout times and updated section 2.3.2 accordingly. The original figure was a composition of different measurements for different elements. We updated Fig. 4 after the repeated experiment (see below). In this experiment we included all relevant elements simultaneously. The example figure shows $^{23}$Na and $^{238}$U.  An average washout time of 0.248 +/- 0.035s at 1% full width maximum for all elements was found. The washout times for all species agree within one standard deviation and there were no longer any large discrepancies found between the elements. We also updated Table 3 accordingly.*

[Figure]

*Fig. 4 revised: New results of the repeated washout experiment for Na and U.*

The calculation of the spatial resolution is important but in spite of making a serious effort, I was unable to understand how this was done. Scan speed seems to have be held constant for different spot sizes and repetition rates. Since a quadrupole ICP-MS was used, was there consideration of synchronization issues, e.g. between acquisition time and repetition rate (van Elteren et al., 2019)? A 1D line profile would be a map with just one line, but can be affected by imaging artifacts nonetheless. This is an additional issue to having a reliable estimate of the washout time, for each element (if different in washout) and ablation parameters used for data production. Then please carefully explain how the resolution was calculated for all settings.

*The calculation of the spatial resolution was done according to the procedure in Sneed et al. (2015), we present here an example calculation and clarified the paragraph (L 181 ff revised manuscript) Low resolution setting: Spot size d: 150 μm, repetition rate f: 20 Hz, scanning speed $v_{scan}$: 40 μm/s, acquisition time $t_{acq}$: 500 ms. The laser fires a 150 μm spot at 20 Hz or 20 shots per second. It moves at 40 μm/s, therefore travels 2 μm per shot. The ICP-MS acquisition time is 500 ms, which equals 10 shots. Adding the distance the laser travels during the washout time therefore the spatial resolution R is:*

$$R = d + \left( \frac{v_{scan}}{f} \cdot t_{acq} \cdot f \right) + (v_{scan} \cdot t_{wash}) = d + v_{scan} (t_{acq} + t_{wash})$$

The experiment with the blank ice is important but needs clarification. Again, one would expect this experiment to be conducted with all relevant ablation parameters that were used for data production, because they are likely to influence sensitivity in analysis. Here the "laser settings for the lowest depth resolution" (what are those?) were used. It would be important to show data for all analytes, not just Na.

*We changed Fig. 5 (see below) and included the results for all elements measured during one blank ice experiment for a 1.3 cm long laser line scan. We included all the relevant laser settings in the text and extended section 3.1 (L 195 ff rev. manuscript) accordingly.*

[Figure]

*Fig. 5 revised: results of MQ ice analysis for all four measured elements*

The initial peak (Figure 5) should not be caused by residual material in the sample line, if the helium is already flowing through the sample lines to the ICP-MS with the laser off?

*We agree, it is much more likely caused by accidentally ablating the adjacent 'glue' (frozen MQ from squeezy bottle) around the sample at the start of the line than by residuals in the sample tubing. We changed the paragraph accordingly (L 198 ff revised manuscript).*

Figure 5 shows clear differences in baseline intensity levels, which are not discussed. Is this due to instrumental drift or, if the setup is used for analysis of other, Na-rich matrices, could this be due to remaining contamination within the ablation chamber?

*The measurements shown in the original Fig. 5 were separated by several months in time in which the baseline intensity levels continuously decreased. The intensity differences are therefore mainly due to gradual deterioration of the optical components within the laser system, which reduced beam intensity over time.*

Relatedly, was there any correction for instrumental drifts, e.g. by ablating a line on the NIST glass standard before and after each acquisition. If not, how were drifts treated for constructing a single dataset from individual lines measured?

*We did not observe any significant instrumental drifts (e.g. changes in ICP-MS baseline with only helium) during the course of each measurement day, usually lasting about 6-8 hours. Therefore we refrained from measuring NIST standards before and after each measurement, because of the risk of contamination and to carry over residual NIST material in the system over to the next sample was considered too high and not beneficial.*

The discussion of the sharp peaks in the blank ice Na data needs to be clarified. If they were noise as proposed by the authors, what is the cause of this noise? They are unlikely caused by the ICP-MS because peaks are absent in gas blank but present when ablating: Evidence for the peaks being related to the ice. The authors speculate that microscopic particles could be the origin, but what about the grain boundaries? Especially relevant for Na. Carefully co-registering the signal in the ICP-MS together with the visual path of the laser crossing a boundary would be immediately helpful to answer this, even if watched by eye only.

*There are occasional sharp peaks in the helium background (see e.g. revised Fig. 5 above), thus we cannot rule out the ICP-MS and persistent residual material from previous analyses as the source of the spikes. We attached an exemplary comparison of a MQ sample picture after measurement with the respective data below. The picture dimension was adjusted to fit the length of the dataset. We cannot find any correlation of the random spikes in the data with the crossing of grain boundaries. Neither for Na nor for other elements. The artificially grown blank ice probably does not exhibit the same grain boundary effects as real glacier ice. Additionally, the two large cracks in the ice which occurred during sample preparation (at ~ 200s and 570s) did not seem to have an impact in terms of larger scale contamination. Additionally, the pre-ablation run would have removed any superficial contamination imposed during preparation. Interestingly the sharp peaks seem to be most present in the Al and Mg data and much less in the Na and especially the Ca. We thus cannot ultimately determine the origin of these spikes and still consider them most likely to be caused by instrumental noise or / and microscopic particles in the ice matrix. We clarified section 3.1 respectively.*

[Figure]

Considering these peaks, how does the blank ice signal look for the "higher depth resolution" settings?

*We only used the low resolution setting for the MQ analyses.*

Regarding a potential cause, could it be contamination or an effect of storage and aging of the ice surface? How long has the MQ ice been stored between preparation and analysis?

*As discussed above, we do not agree that storage time alone would lead to increased contamination of the sample surface. The MQ samples were prepared along with the real ice samples and also stored no longer than 3 days at maximum. No deterioration of the surface by sublimation could be observed after this time.*

Maybe the occurrence of these peaks could be avoided by decontaminating the sample just before analysis.

*Again, as discussed above, we do not agree. The samples have a perfectly smooth and clean surface after the final microtoming step and an additional manual scraping directly before analysis would destroy the smoothness of the surface along with the risk of introducing additional contamination. Any superficial contamination that would have been deposited on the surface after preparation would be removed during the routinely carried out pre-ablation step.*

In line 192, the authors write "*The average intensity of the signal retrieved from the blank ice is barely distinguishable from the one with just helium*" and then in line 196 onwards "*The median of the Na signals from MQ ice is on average about 9% higher than the one of the helium stream*", which is contradictory.

*We agree and clarified the paragraph (L 199 ff revised manuscript).*

Ultimately, it cannot be ruled out that the peaks are related to remaining contamination on the ice, which is detectable due to the high resolution of LA-ICP-MS analysis. The removal as outliers is not convincingly justified. Visual images taken during ablation could help to test the hypothesis that these features are related to the ice.

*We attached a picture of a MQ sample after analysis above, but cannot find any correlations between the visual features of the ice and the spikes in the laser signal.*

The second area concerns signal interpretation as evidence for stratigraphic (annual) layering and the comparison with continuous flow analysis (CFA). The following points need to be addressed and clarified:

To my understanding the manuscript does not show any raw data, but only smoothed datasets. There should be generally showing of raw data (for multiple elements) and the effect of the applied smoothing.

*We agree in principle, however in terms of readability it did not seem meaningful to show the raw data because of the noise and high frequency signal masking the important large scale features that were discussed. We adapted Fig. 6 and 7, now showing the unsmoothed datasets in the background (see below)*

[Figure]

*Fig. 6 revised: overview of the LA-ICP-MS and CFA measurements for the shallow core section: Raw data as light colour in the background*

If the goal was to identify annual layers in the LA-ICP-MS line profiles, one would have expected to see data covering a section where CFA is able to clearly resolve them.This data exists for the Skytrain ice core (Hoffmann et al., 2022).

*In Fig. 6 we show a section of the core where this is indeed the case.*

There, the LA-ICP-MS vs CFA comparison could show that i) annual layers exist for the investigated element and ii) provide guidance on how to identify them in the LA-ICP-MS data, in particular which elements, and ultimately the demonstration that annual layer detection is possible with this LA-ICP-MS data. I think the authors have tried something similar but am not certain. Figure 6 shows an exemplary comparison between LA-ICP-MS and CFA data for a section between 83.2 and 84 m depth. On the one hand, this is below the depth of 60-70 m until which the identification of annual layers in Na was possible in CFA (Hoffmann et al., 2022). However, the authors write later in line 275 "*The age model in this core section is based on annual layer counting of the CFA Na and Ca signals constrained by absolute age markers, mainly volcanic eruptions at this depth*" – so was it possible to count annual layers at this depth in CFA? If so, where are they in Figure 6?

*The annual layer identification based on CFA data in this depth was possible and was done based on a combination of the Na and the Ca signal as elaborated in Hoffmann et al. (2022.) We added the position of the identified layers in Fig. 6 as black lines (see above).*

*We clarified the first part of section 3.2 with respect to the layer identification.*

On the other, the manuscript states that the expected annual layer thickness based on the age model at this depth is 11 cm, which should be within the range of Na depth resolution (3.8 – 4.7 cm, Hoffmann et al., 2022) and, the CFA Na data in fact shows 7-8 peaks within 80 cm of Figure 6.

*The Na CFA data shows 6-7 peaks in this selected depth interval, therefore the resulting annual layer thickness based on CFA Na alone is 11-13 cm in this depth interval.*

Yet the authors state in line 220 "*This finding indicates, that even at this shallow core depth the sodium CFA signal might not be able to resolve seasonal variations present in the ice.*" and in line 228 "*This is surprising because, according to the spatial resolution of  4 cm of the CFA ICP-MS, it should in principle be possible to identify features smaller than the roughly 10 cm variations that are visible in the Na data. We attribute this remarkable smoothing of the CFA data mainly to turbulent mixing effects in the sample lines and the ion source of the CFA ICP-MS.*". This would mean that annual layers remain obscure at this depth 83.2 – 84 m and that the CFA resolution previously reported in Hoffmann et al. (2022) was overestimated? I find this confusing and tried to highlight the ambiguities.

*We agree that some of the highlighted statements above were misleading and clarified the first paragraph of section 3.2 with respect to the annual layer discussion. We did not intend to give the impression that the previously reported CFA resolution was underestimated but to emphasize the capability of the laser to reveal smaller scale variations.*

Figure 8 is related to this issue, but shows fluorescence Ca from CFA, which has a higher depth resolution of 1.4 cm (Hoffmann et al., 2022). In Figure 8 it is compared to LA-ICP-MS Na, however. I would expect to see the direct comparison to LA-ICP-MS Ca which exists in Figure 6? Notably, also in Figure 8, CFA Ca shows about 8 peaks within 83.2 – 84 m depth.

*We revised Fig. 8. It now shows the Na LA-ICP-MS signal compared to the respective CFA signal along with the spectral analysis of both datasets (see below).*

[Figure]

*Fig 8 revised: Left: Na LA-ICP-MS signal compared to CFA signal. Right: Spectral analysis (multitaper method) of both signals on the left*

To reiterate, showing the raw LA-ICP-MS data together with the smoothing is important. At present the smoothing levels appear arbitrary. E.g. in line 235 the authors write "The

theoretical expected annual layer thickness at this depth is about 2.1 mm and therefore much smaller than the resolution of the CFA data. Again, to increase readability the LA data were smoothed using a running average of 4.5 mm." This would mean you are eliminating a potential annual layer signal by smoothing, which could not have been what you intended?

*The level of smoothing, which was chosen for reasons of readability might have obscured relevant features. We changed Fig. 7 (see below) and only applied a smoothing in the range of the spatial LA-ICP-MS resolution of 185 µm.*

[Figure]

*Fig. 7 revised: deep section of Skytrain ice core ~ 20 kaBP. Raw data added as light colour in the background.*

The reasoning that spectral analysis can reveal periodic signals in the LA-ICP-MS data that correspond to annual layer predictions by the present age model is not convincing due to the following reasons: a) the annual layers are unlikely periodic over small depth intervals. The reality is much more complex, some layers can be spaced at smaller, some at greater distance (some could be missing entirely).

*Yes. We are aware of the complex nature of annual layering in ice cores. However, in the particular shallow depth interval shown in Fig. 6, the spacing of the CFA Na peaks appears to be very regular. This depth interval therefore serves as a model section to test the capabilities of the spectral analysis for deeper sections. The intention of this analysis is to discover layering as defined in the manuscript (L 25ff), which is not necessarily annual but characterised by periodic changes in impurity concentrations in general. We thus consider spectral analysis a useful tool to reveal periodicities that are preserved in the ice which*

*cannot be resolved by the CFA. In the deeper sections, where layer thicknesses are theoretically in the mm range, 40 cm of ice encompass several hundred years, which is sufficient for a reliable analysis of periodicity.*

The mentioned peaks in Figure 8 are illustrating this uneven spacing. b) the age model predictions (especially in the deeper parts) can have large uncertainties (which are not quantified in the text, however), c) it is not clear if the annual layers have even survived interactions with the ice matrix, e.g. given the potential imprint of grain boundaries on the Na signal.

*We agree that it is unclear to what extent the annual layers are preserved in the deep ice, although there are indications that this can be the case despite the effects of crystal growth and movement (Svensson et al. 2011). The main goal of this study is not to only detect* **annual** *layers in Skytrain ice core but to use the LA-ICP-MS technique to reveal quasi-periodic layering over longer timescales (e.g., decadal, centennial) which cannot be resolved by the CFA.*

An in-depth test case from more shallow depth with annual layers in CFA and some indication on the imprint of the ice matrix (optical laser camera) would help to establish this approach, because at present it is not justified enough. The ambiguity regarding deep ice conditions remains regardless of this, however.

*The test case at a shallow depth is presented and discussed in sections 3.2 and 3.3.1. A picture with clearly visible grain boundaries and ice matrix structure is already presented in Fig. 2c.*

The authors are aware of these limitations and write in the conclusions "*Nevertheless, the frequency analysis cannot conclusively and unrelated to other information be used to identify horizontal layering.*", but then one wonders why no other information was used here? The same goes for a more extended use of the parallel line approach. In one case, 8 parallel lines were measured and compared. Figure 10 shows the results, which are encouraging, but the "clear correlation" with a sine wave is an overstatement. There is one section with higher Na intensities and one section with lower intensities, at the bottom and the top section of the 8 lines (Figure 10, left). A longer profile would have been needed to discuss signal periodicity.

*We updated Fig. 10 (Fig. 9 revised manuscript) and removed the potentially misleading sine function. We agree that it would have been desirable to perform the parallel line experiment on a longer ice section, but because of time constraints it was not feasible at the time of analysis. We intend to intensify this kind of analysis in the future. However, we find that the general course of the stacked signal exhibits a remarkable similarity to the expected annual periodicity, even on this short piece of ice.*

A second case with 3 parallel lines is mentioned but the original profiles not shown. I would strongly recommend to extend this type of experiment with parallel lines, because it could bring much more clarity to the data especially in the deep ice. Note that I am not asking for full 2D imaging, which was clearly not within the scope of this work or maybe not possible.

*2D imaging has successfully been done with this setup, but was not the focus of this study. We changed section 3.4 entirely, and removed the results of the wavelet analysis, because it*

*might have been confusing and misleading. We extended the discussion of the 3 stacked lines with respect to grain boundary effects.*

The importance of measuring parallel lines has been shown before (e.g. Della Lunga et al. (2017) stated that "the averaging of the LA signal between two or more parallel tracks spaced by a few millimetres is not only desirable, but necessary"). The subsampling of 2D images done by Bohleber et al. (2021) showed that the grain boundary imprint can remain dominant to parallel lines up to a resolution of a few 100 microns.

*We are aware of the previous studies on parallel lines, thank you. However, the choice of distance between parallel lines strongly depends on the grain sizes, impurity concentration and ice core specifications. And to repeat, there are indications that despite the influences of grain boundaries, layered structures can still be preserved in deep ice (Svensson et al. 2011). There are hints that at least in the sodium signal this is indeed the case in Skytrain ice core, which is corroborated by the spectral analysis (see section 3.4 revised manuscript and revised Fig. 12 below).*

For the parallel line experiment in Figure 10 the data was filtered according to the "*theoretical depth resolution of 185 μm*". Some high-frequency variations remain in the stack. The authors write in line 300: "*These small scale variations can be caused by (i) incomplete decontamination of the sample surface, (ii) particles that show up as sharp peaks in the ICP-MS signal (iii) actual small scale intensity variations e.g. caused by accumulation of Na in crystal grain boundaries. Based solely on the line data presented here, which of these possible sources is the most dominant cannot be determined.*" This could be very problematic, because it would mean that it cannot be ruled out that the signals are influenced by contamination? (See comments on blank ice above).

*Contamination is very unlikely (see comment and blank discussion above). The grain size in this core section is generally very large, but with a high variability (~ 4 +/- 3 mm), therefore grain boundary effects together with mobilisation of particles are considered to be the most likely reasons for the high frequency variations. We clarified section 3.3.2 respectively.*

Section 3.4 greatly suffers from the initially mentioned remaining ambiguities in the data and needs to be revised carefully.

*We changed the section and now exclusively focus it on the in-depth discussion of the ice core section from ~ 26kaBP (see revised Fig. 10 below).*

[Figure]

*Fig. 10 revised: deep section of Skytrain ice core ~ 26 kaBP. CFA data for comparison in black. Raw data added as light colour in the background.*

[Figure]

*Fig. 12 revised: Results of the spectral analysis of the long profiles (left) shown in revised Fig. 10 (above) and the stacked lines (right).*

In Figure 11, it is not clear how much smoothing was applied, but in any case, the low-pass filtered signal suggests close to twice as many maxima as the sine curve. It is unclear to the reader what these maxima may mean and it is unclear what uncertainty the "theoretical annual layer thickness" has for this 10 cm section. Then line 322 reads "*Comparing the 2-4 mm periodicity to the sample picture in Fig. 11, suggests that this periodicity is likely to correlate with the estimated average crystal grain size in this depth. The 2-4 mm periodicity therefore*

*probably does not represent layering in the sense of larger scale horizontal features, but accumulation of sodium in the crystal grain boundaries.*" None of this is actually shown. Again, in the absence of imaging the parallel line approach could shed light on the grain boundary imprint, especially if combined with optical images of the laser camera. Notably this imprint could vary distinctly between sections and ice conditions and hence it is not sufficient to discuss this exemparily.

*We added a new Figure (Fig. 11 in revised manuscript), comparing the optical image of the sample to the stacked signal of the 3 parallel lines (see below). The influences of the grain boundaries (highlighted by green vertical lines) remain complex. In this particular section, the crossing of grain boundaries by the laser path (yellow lines in Fig. 11) much more correlates with sharp peaks in the Mg and Al signal than it does with the Na signal.*

[Figure]

*Fig. 11 revised: Comparison of sample image to signal of stacked laser lines (yellow) for a deep section of Skytrain ice core. The positions of the crossed grain boundaries are highlighted in green.*

Just one other example of the ambiguity is the discussion of Magnesium in line 334 onwards. After noting detected areas of enhanced power in the spectral analysis, it is stated that "*Magnesium is known to show a similar seasonality as sodium (e.g. Curran et al. (1998))*." However, Figure 6 shows no evidence of this potential seasonality. Then it is noted that "*During the glacial, the total magnesium content contains a large fraction originating from mineral dust particles (de Angelis et al., 2013), which is not the case for sodium. The maxima in the fourier spectra at very small wavelengths of about 250 μm could hint to effects generated by ablation of such single particles. The soluble magnesium fraction can migrate into crystal grain boundaries (e.g. (Bohleber et al., 2021)). We therefore attribute the larger periods (3-6 mm and 8-12 mm) again to grain boundary effects.*" The last sentence is speculative and not supported by data. Observing the laser cross grain boundaries and simultaneously recording Mg intensities would have provided important information here.

*We extended the discussion of the grain boundary effects for all elements (see response above) and changed section 3.4 respectively.*

The text goes on to say that "*It is remarkable, that the magnesium data shows an additional area of enhanced, although as well strongly intermittent power, around ∼ 1.5 mm period length and thus in the range of the annual layer thickness from the age model. In summary, these findings could be an indication that the magnesium is less affected by grain boundary effects than the sodium. In consequence, magnesium might be better suited for identification of annual layers, but this needs to be confirmed by more detailed analysis (parallel lines and signal stacking) in the future.*" Also here, no proof for these statements referring to Mg being a better annual layer indicator are presented in the data.

*We revised section 3.4 entirely. The original version might have suffered from overinterpretation of the spectral and wavelet analysis. We adapted the discussion to a more general analysis of the features preserved in the different species.*

This illustrates how the manuscript suffers from not having established well how an annual layer looks like in the different chemical species observed by LA-ICP-MS and discussing more thoroughly the differences between the chemical species shown in Figure 6. Based on CFA in Figure 8, Ca shows the most promising annual signal, in line with the findings by Hoffmann et al. (2022), although a better comparison with LA-ICP-MS Ca should be made in Figure 8. The "striking alignment" between CFA Ca and LA-ICP-MS Na in Figure 8 is an overstatement. Unfortunately, there is little Ca data presented due to a technical issue, line 241: "The LA-ICP-MS calcium signal suffered from a contamination issue in the ICP-MS at the time of these measurements." How could a Ca-specific contamination in the ICP-MS have happened? This needs to be explained. Later on in the conclusions, the authors write "The calcium signal however was lost in the noise for most measurements."

*As stated above, the Cambridge LA-ICP-MS system is not exclusively used for ice but for geological samples as well. The overall backgrounds especially in mineral relates elements might thus be higher than in other laser systems only dedicated to glacier ice analysis. The specific Ca contamination happened during laser ablation analysis of minerals on the previous day. Sodium was the main target of our analysis, therefore and with respect to the limited time at the instrument, the enhanced Ca background was accepted at the time.*

With all this taken into consideration, it is strange that no parallel lines were measured for the deepest ice samples with arguably the most complex conditions, line 362 onwards. This section is again mostly speculative about the origin of the features in the spectral analysis and needs better support by data.

*We removed the single line discussion of the deepest sample. The sample originated from a disturbed ice section and was not representative for the general LIG conditions. The assessment of the capabilities of the analytical system, which was the main focus of this study, can be sufficiently discussed using the samples from 26 kaBP.*

The conclusions need to be carefully rewritten with special emphasis on replacing speculative statements with statements that are actually supported by the results.

*We revised the conclusions accordingly.*

Specific comments

Considering the amount of major comments, I am only including a few specific comments below. This mostly concerns how references are made to existing literature, which should be extended and placed more carefully.

- Line 155: This needs to be rewritten after a more careful assessment of the washout times, although it is not entirely clear what the authors intend to show with the comparison of washout times to previous studies. Consider that the washout time as single pulse response is determined as FW0.1M or FW0.01M, and not by a single flank of the pulse signal. Otherwise you cannot compare these values to other studies.

*We corrected the section accordingly (see reply above).*

- Line 349: Regarding the role of Aluminium and its relationship to particles, the actual relationship between a specific element and features like particles needs to be shown first and should not be assumed a-priori, e.g. some recent data by Bohleber et al. (2023).

*We changed the paragraph.*

- This is not the first study showing that heavily smoothed LA-ICP-MS signals agree with what is measured by CFA. The LA-ICP-MS vs CFA agreement has been shown and exploited in previous works (Sneed et al., 2015; Della-Lunga et al., 2017; Spaulding et al., 2017). The previous studies need to be credited accordingly in the respective statements in the text, of which there are several instances (e.g. lines 226).

*We added respective references.*

References

Bohleber, P., Roman, M., Šala, M., Delmonte, B., Stenni, B., & Barbante, C. (2021). Two-dimensional impurity imaging in deep Antarctic ice cores: snapshots of three climatic periods and implications for high-resolution signal interpretation. The Cryosphere, 15(7), 3523-3538.

Bohleber, P., Stoll, N., Rittner, M., Roman, M., Weikusat, I., & Barbante, C. (2023). Geochemical Characterization of Insoluble Particle Clusters in Ice Cores Using Two-dimensional Impurity Imaging. Geochemistry, Geophysics, Geosystems, 24(2), e2022GC010595.

Della Lunga, D., Müller, W., Rasmussen, S. O., Svensson, A., & Vallelonga, P. (2017). Calibrated cryo-cell UV-LA-ICPMS elemental concentrations from the NGRIP ice core reveal abrupt, sub-annual variability in dust across the GI-21.2 interstadial period. The Cryosphere, 11(3), 1297-1309.

Hoffmann, H. M., Grieman, M. M., King, A. C., Epifanio, J. A., Martin, K., Vladimirova, D., ... & Wolff, E. W. (2022). The ST22 chronology for the Skytrain Ice Rise ice core–Part 1: A stratigraphic chronology of the last 2000 years. Climate of the Past, 18(8), 1831-1847

Müller, W., Shelley, J. M. G., & Rasmussen, S. O. (2011). Direct chemical analysis of frozen ice cores by UV-laser ablation ICPMS. Journal of Analytical Atomic Spectrometry, 26(12), 2391-2395

Sneed, S. B., Mayewski, P. A., Sayre, W. G., Handley, M. J., Kurbatov, A. V., Taylor, K. C., ... & Spaulding, N. E. (2015). New LA-ICP-MS cryocell and calibration technique for sub-millimeter analysis of ice cores. Journal of glaciology, 61(226), 233-242.

Spaulding, N. E., Sneed, S. B., Handley, M. J., Bohleber, P., Kurbatov, A. V., Pearce, N. J., ... & Mayewski, P. A. (2017). A new multielement method for LA-ICP-MS data acquisition from glacier ice cores. Environmental science & technology, 51(22), 13282-13287.

*Svensson, A., Bigler, M., Kettner, E., Dahl-Jensen, D., Johnsen, S., Kipfstuhl, S., Nielsen, M., and Steffensen, J. P.: Annual layering in the NGRIP ice core during the Eemian, Clim. Past, 7, 1427–1437, https://doi.org/10.5194/cp-7-1427-2011, 2011.*

van Elteren, J. T., Šelih, V. S., & Šala, M. (2019). Insights into the selection of 2D LA-ICP-MS (multi) elemental mapping conditions. Journal of Analytical Atomic Spectrometry, 34(9), 1919-1931.

---

## Author Comment (AC2)

**Replies to reviewer 2 comments in calibri italic**

*Thank you for your review and your valuable input. We agree that there were some technical inaccuracies and the signal interpretation including especially the CFA – LA-ICP-MS comparison section needed some clarification.*

General comments

Paper by H.Hoffmann et al. presents the laser Ablation - ICP-MS measurements for high resolution chemical ice core analyses with a first application to an ice core from Skytrain Ice Rise (Antarctica). The new Cambridge LA-ICP-MS ice measurement system was applied to selection 130 samples over 6.7 m of the Skytrain Ice Rise ice core. Authors claim that that periodic concentration changes on the millimetre scale can be identified in ice from 26 ka BP.

LA-ICP-MS is a relatively new, promising and challenging technique that offers ultra-high resolution data. Clear method descriptions and in-depth discussions are essential for such studies, as some challenges, such as elevated concentrations at crystal boundaries and contamination, can only be addressed experimentally.

The section on ablation settings and resolution needs further clarification.

*We changed the section and removed a few small errors.*

It appears from the text that the settings were optimized in each measurement session, but it's not clear how many measurement sessions there were. It seems from Table 4 that this was based on depth/age-dependent resolution requirements, but the text states that it depends also on ice sample surface conditions.

*This statement was misleading. The surface conditions of the ice were the same for alle samples, but the other parameters (expected layer thickness impurities etc.) were not. We corrected the first paragraph of section 2.3.3 (L162 ff revised manuscript) accordingly.*

The adjustments made for each sample and why they were made is not clearly outlined.

*The settings for the glacier ice sample measurements were optimised at the beginning of every measurement session, which usually lasted 3-4 days with several weeks to months in between. There were 10 sessions in total. During this time, the laser ablation system was not only used for ice but for geological samples and was subject to the normal processes of usage and wear. This led to the need of readjustment of the major settings (laser power, repetition rate) for each measurement session. Within the session, the settings were kept constant, to ensure that each continuous set of 8 or 16 consecutive ice core samples were analysed with the same parameters. We amended section 2.3.3 accordingly (L 165 ff revised manuscript).*

Regarding the low-resolution value mentioned in the text (182 µm) versus Table 4 (185 µm), there seems to be a discrepancy.

*We corrected the numbers.*

Were any images taken during ablation? This could help understand the source of noise and peaks detected in MQ ice.

*Yes there were images taken before and after measurement of each sample. We can provide an example picture from the on board camera in the appendix (see below, 2 x 3.8 mm, ablation path on the left), but we cannot see an advantage of this kind of picture compared to the high resolution photographs (2 x 5 cm example picture below). The image of the MQ sample did not reveal any features, which could explain the peaks in the signal*

[Figure]

[Figure]

*Mosaic on-board camera picture (2x3.8 mm)s from the Cambridge laser system on top. Picture of a 2 x 5 cm sample from ~ 100 ka BP from Skytrain ice core on bottom, grain boundaries and bubbles clearly visible*

My major concern is the annual dating section and comparison to CFA. It's indeed an important point. It would be better to see the initial raw data and then the results of different smoothing techniques. Some statements in this section are not fully supported by the data/figures.

*We revised Fig. 6 and 7 and added the raw data in the background. We also revised the discussion to be more precise about the detection of features.*

L.215-217 should be included in the figure caption.

*Done.*

The statement at L.220-221 about the higher frequency fluctuations detected by LA-ICP-MS compared to CFA is somewhat subjective and depends on the criteria used for annual counting. It's not clear how annual layers were identified before at this depth using CFA. Could you plot annual boundaries?

*We changed Fig. 6 accordingly, added the annual layer positions and extended the discussion of the layer detection.*

L.222 Regarding the statement about the correlation being less obvious but still visible, it might require further explanations or additional smoothing. At some places, correlation seems to be negative. The dip in concentration around 83.7 m is not well seen in the same place in LA-ICP-MS data. Further smoothing of the LA-ICP-MS data could make it visually comparable however, the raw data should be provided as well.

*We reformulated the paragraph and weakened the statement about the magnesium correlation.*

From Section 3.2, I'm not convinced that the conclusion holds true for all the elements mentioned at L.248. L.260, why compare calcium to sodium and not to 43Ca measured with LA-ICP-MS? The "striking alignment" mentioned might be overstated.

*We changed section 3.2 and Fig. 8 (see below). We now only compare the Na LA-ICP-MS signal to the respective CFA signal. The laser Ca background was generally too high and therefore the signal mainly below detection limit*

The section on spectral analysis is not very convincing since although overall periodicity for annual layers may be expected, we are looking at 1 m of the core and annual layers of approximately 10 cm. It's just ten years, and in this case, you can simply plot suggested boundaries and discuss discrepancies in CFA vs. LA-ICP-MS and differences for specific measured species.

*This is what we did in section 3.2. We agree that the spectral analysis would not have been necessary in this shallow section. This depth interval with well known annual features serves as a test case for the spectral analysis method to be applied in deeper sections later.*

Section 3.3.2 Needs clarification and possibly extension.

"Based on the conclusion here, which of these possible sources is the most dominant cannot be determined." I would have expect similar experiment for other sections of the core with different grain size and different annual layer thickness in order to assess. The sources and level of contamination should be explained in this regard too.

*We agree that this statement was not precise and misleading. We reformulated the paragraph.*

At L.291, why was this section of the core chosen for the experiment? Did you perform line staking for other sections? Please comment and explain in the text.

*This section was chosen, because the expected layer thickness is below the resolution of the CFA but still large enough to be not significantly affected by grain boundary effects. We changed the beginning of section 3.3.2 accordingly (L305 ff). Yes, we also performed parallel line measurements for other depth intervals. One deep example is discussed in depth in section 3.4- The ice section discussed in 3.3.2 was chosen because the expected annual layer thickness is below the resolution of the CFA, but still large enough to probably not be affected by the grain boundary effects too much.*

Section 3.4 is rather speculative at the moment, which is also stated by the authors. I would recommend shortening it and reporting the main findings, with most of the Wavelet spectra and Fourier spectra put into supplementary materials since they show similar features.

*We changed 3.4 entirely. We removed the results of the wavelet analysis, which might have been confusing and reduced and clarified the paragraph with respect to the discussion of the section from 26 kaBP only. The section is now focused on the investigation of signal preservation together with the influences of the grain boundaries.*

---

## Author Response (AR2)

**Replies to reviewer 1 comments in calibri italic**

General comments

I appreciate the effort that the authors have put into the revision of their manuscript, which has improved significantly. Many aspects are now more comprehensible, not least to the restructuring of the text and improved figures. Aside from minor and technical comments that were omitted during the first round of review, a few important points remain that need to be clarified for the reader.

*Thank you for your review and input, we tried to address all issues in the revised version and will clarify the remaining points within this round.*

Presentation of the instrumental setup

The presentation of the new cryo-LA-ICP-MS setup has been improved and includes now several important pieces of information. At least one optical mosaic that the authors show in the response should be included in the main text, e.g. adding to Figure 3. It is important to see the ablated laser path in this detail.

*We already provided a close-up of the laser path in the revised Fig. 3. We added an overview mosaic picture in the appendix (Fig. A1) to not disproportionately extend the main manuscript.*

To provide an example of why this is important: With regards to the difficulty in achieving ablation, the authors write in their response "This might be due to the finely microtomed and thus much smoother, more reflective and less uneven surface of the samples compared to other systems." First, the overview pictures do not allow to judge surface roughness adequately, e.g. in the comparison included in the response using the image taken from Bohleber et al. (2023). The latter was also prepared by a machined microtome prior to cryo-Raman analysis at which point the overview image was taken (Stoll et al., 2022).

*We agree that this particular picture comparison might not have been the best choice to judge the surface roughness. However, over the course of three years of laser ablation measurements on many different samples we found some empiric evidence for the correlation between surface roughness and ablation efficiency. This is not yet quantifiable.*

Second, there is no evidence for the surface roughness increasing ablation and, in a purely speculative way, one could also argue in the opposite direction: More roughness leading to the laser being out of focus more often, etc.

*We agree that there is no theoretical evidence for the efficiency of ablation depending on the surface roughness. Roughness on a scale that would lead to the laser being out of focus would indeed result in lower ablation efficiency. But roughness on a scale below the laser spot size could have the opposite effect.*

Ice is highly transparent at 193nm, the surface thus hardly reflective. At present the high fluence needed remains enigmatic but shows how little is known about the laser-ice interaction at UV-wavelengths.

*Not just the laser-ice-interaction in general but also the apparent dependence on the different ice features (impurity concentration, opacity, crystal size and orientation (?), bubble content etc.). This also shows the need to compare more laser ablation measurements across different systems to better understand the mechanisms of ice ablation.*

We can probably agree that this is interesting and hopefully it can be cleared up in the future with more dedicated experiments.

*Agreed. Future experiments to investigate this phenomenon are planned.*

The difficulty in coupling leads to the need to adjust the ablation settings, including fluence and repetition rate. The authors argue that this is caused by the eventual wear and tear of the laser optics. It should be clarified in the text if the fluence was actually measured at the sample level, where this degradation of the optics would manifest, or if the fluence reported corresponds to the nominal fluence settings in the software. Table 2 says "laser fluence on target" – was this measured at the target? Otherwise this should be "nominal laser fluence".

*The fluence was not measured directly at the target but refers to the nominal laser fluence reported by the software. The term in Table 2 and throughout the manuscript was corrected accordingly.*

The mosaic shows much better than the present close up in Figure 3 how deep the ablation craters are – and shows that they are consistent along the line as far as one can tell. It would be of interest to include the ablation settings used alongside this image, especially the repetition rate, which increases the overlap of spots fired and hence the ablated volume or crater depth.

*We included the respective settings in the figure caption of Fig. A1. They were: Scanning speed: 50 $\mu ms^{-1}$, nominal Laser fluence: 6.7 J cm$^{-2}$, Repetition rate: 120 Hz. Spot size: 150 $\mu m$ round*

The experiment for determining the washout time (or single pulse response, SPR) now nicely illustrates the performance of the system. However, as remarked in the first round, it is puzzling why the ablation settings were again different from the settings actually used in the experiments on ice?

*We did not use the same settings as for ice on the NIST standard because this would have caused way too much material being ablated from the glass standard and therefore the ICP-MS signals going into saturation. The results would not have been meaningful.*

Compared to ice analysis in the SPR experiment fluence was lower, 4 Jcm-2 vs 6 – 6.7 Jcm-2 , and spot size smaller, 30 μm vs 150 μm. It has been shown that both parameters matter for determining the SPR (e.g. Jerše et al. 2022). More material will generally need more time for transport, and a 30μm spot ablates only a small fraction of the area of a 150μm spot.

*We agree that there might be a general relation between peak shape and beam size. However, Jerše et al. 2022 state that the appearance of double peaks is additionally strongly dependent on the material matrix and the generation of large particles. The matrix of the NIST standard is very different to glacier ice. The effects on the SPR are not quantified in Jerše et al. 2022. Since in this study it was not possible to perform the SPR experiment on a homogeneous ice matrix standard, we argue that the NIST experiment with the reported settings gives the currently best estimate for the SPR of our setup. We added a paragraph (L 161 ff revised manuscript) weakening the statement about the determined washout time.*

The SPR value is further on used to determine the spatial resolution. This is an important number that appears in several central places throughout the manuscript. The authors also write in line 177 "Therefore the acquisition time is the dominating factor regarding depth resolution compared to the much smaller washout time." – this can only be stated if we know the SPR for 150μm.

*As stated above, we can only use the results SPR determined with the 30 μm spot size and 4 Jcm$^{-2}$ as a best estimate. The acquisition time of the ICP-MS is with 500 ms almost double the measured SPR times (198-270 ms). Even if the peak shape would change for the ice ablation, we doubt that this would cause an increase of the SPR by a factor of two or more.*

This is an important issue that needs to be addressed, either via a clear explanation to the reader that the SPR for 150μm remains unknown and likely underestimated by the 30μm experiment or preferably by determining this value experimentally.

*We recognize that it cannot be proved entirely that the acquisition time is larger than the washout time, but we still consider it very likely (see comment above). We changed the sentence in line 177 (L 185 ff revised manuscript) accordingly.*

Please also adjust the statement in the abstract which reports "depth resolution of down to 80 μm".

*Done. Changed to 182 μm*

The spikes in the background signal and their subsequent removal remains an important issue to clarify further, especially with respect to the discussion about a potential imprint of grain boundaries (see below). The authors write in line 209: "The origin of the sharp isolated, high peaks is not entirely clear." And then "Therefore, we conclude that the very high frequency signals and sharp peaks in intensity are most likely caused by electronic interference and should be regarded as analytical noise." – There is no evidence provided for thsis effect being electronic interference to support this conclusion. More importantly, based on the statement in lines 215 – 217, it is not fully clear if peaks were removed from the actual data acquired on ice. If so, how would this affect a localized high intensity spike potentially caused by a grain boundary?

*We reformulated the paragraph (L 216-223 rev. manuscript) and clarified the statements. The sharp, high, single point excursions in the data were regarded as noise and removed from the data during background correction (L 221f rev. manuscript). We found that the crossing of grain boundaries causes peaks much wider than just one datapoint (see discussion Fig. 11). The identification of these features should therefore not be affected by the background noise removal.*

Layer detection in the Skytrain ice core

If a seasonality is established for LA-ICP-MS Na it makes sense to look for such periodicities in deeper ice, using spectral analysis if not evident otherwise. However, the imprint of grain boundaries needs to be demonstrated more clearly than what is presently discussed around the new Figure 11. It is generally important to clarify how the authors regard the high frequency signal components in their data. Are they considered noise, like for the blank ice, or are they a signal that has a physical origin in the ice, e.g. from grain boundaries?

*As stated above, only the very high, single point excursions were regarded as noise and removed from the data. Still, some high frequency variations in the depth domain remain after this outlier removal.*

Figure 11 relates to the aforementioned issue of outlier removal. In the data shown here, were high peaks removed as outliers as for the ice blank data?

*Yes. As described above, the very high single point outliers were removed from the data for all samples.*

In line 304 the authors write "The main challenge of identifying small scale features in the LA-ICP-MS data, which can be interpreted as layers, is to separate the signal of interest from the high frequency, noisy background." – this statement suggests that the noisy background illustrated for the blank ice still exists?

*This statement was misleading, the background removal of high single, spikes was done for all samples before further (e.g. spectral) analysis. We reformulated the sentence (L 309 rev. manuscript).*

Then, in line 307: "When the laser ablation path crosses a grain boundary, the enhanced concentrations can cause sharp peaks in the LA-ICP-MS data which could then lead to misinterpretation of grain boundaries as larger scale features." – this statement argues that there is a physical origin of these signals.

*As stated above, after background removal of the single datapoint excursions, some sharp peaks and steep excursions remain. The sentence was reformulated (L 310 rev. manuscript).*

And in line 328: "Superimposed to the large trend the smoothed signal still shows high frequency variability. These small scale variations are most likely caused by either (i) particles that show up as sharp peaks in the ICP-MS signal (ii) actual small scale intensity variations e.g. caused by accumulation of Na along crystal grain boundaries."
The latter statement sounds similar to line 307 but is speculative, because there is no direct evidence for (i) or (ii). Figure 11 is interpreted as little evidence for Na at grain boundaries (line 370, 399).

*We reformulated the respective paragraph and emphasized the hypothetic nature of the statement (L 333 ff rev. manuscript)*

I find the hypothesis that localization of elements at grain boundaries – if present – produce detectable signals in the frequency domain still not convincing. Grains would need to be highly uniform in size to generate periodic signals, which is not the case – all visual images included in the manuscript show highly variable grain sizes. This needs to be addressed better and rephrased (e.g. line 430).

*We agree that there is some variation and range in the grain size. And that it is unlikely that a certain grain size will show up as frequency in the PSD. However, we know about the effect of impurity concentrations in the boundaries and that this will have subsequent impacts on the overall impurity distribution of the sample. This will in return influence the periodicity. We therefore still regard the ranges of the PSD that cover the span of the grain sizes as influenced by these effects. We added a table documenting the average grain sizes and ranges for all samples in the Appendix (Table C1). We reformulated the statement in L 434 rev. manuscript*

I suggest removing indicating the grain size in the PSD plots, as this suggests that such periodicities could be expected.

*We would not like to remove the grain size ranges from the PSD plots. They function as a guidance for the reader indicating the mean and range of the grain sizes and therefore highlighting the period lengths that might be influenced by these effects. We do not discuss direct connections between the periodicities and the grain sizes, only ranges.*

The parallel line approach is much more convincing for reducing grain boundary related signals. (Figure 9). This is why I find it confusing that in Figure 11, three lines were stacked specifically to discuss grain boundary imprints. Stacking would make the grain boundary signals much weaker, not stronger. How does this comparison look for the single line profiles?

*We used the stacking method because the three lines were very close together, crossing the same grain boundaries at almost the same positions. We hereby expected the stacking to enhance the common features, but that might have been misleading. We changed Fig. 11 and now only show a single line comparison and additionally the two other lines in the Appendix (Fig. B2).*

In their response the authors write that grain boundaries are observed not wider than 5μm. At a resolution of more than 180μm (but see comment on SPR above), the signal contrast caused by a grain boundary could be too weak to detect, depending on the actual impurity concentration in the boundary. This point could be worth mentioning.

*We agree in principle. Nevertheless, we observed the same phenomenon (some grain boundaries showing a very distinct signal, some showing none) during 2D mapping (data not shown in the paper) with much smaller spot sizes (50 μm) and larger scanning speeds (100 μm/s). Therefore we do not believe that the larger spot size combined with a slower speed and thus longer dwell time on the grain boundary would smooth out the signal completely. It would be interesting to analyse ice samples from different origins, impurity contents, microstructure on the same laser ablation system, to investigate if this is an observation unique to Skytrain ice samples. Such analyses are planned for the future.*

Line 235 and Figure 6: Please explain how the grey vertical lines for the annual layers were identified in the CFA. The text mentions CFA-Ca, but in Figure 6 only Na shows a clear oscillating pattern. Interestingly the correlation with Ca is weak at best, e.g. the big peak around 83.8 m was skipped in counting and at 83.95m a Ca peak is missing but a grey line is set?

*As elaborated in Hoffmann et al. 2022 the annual layer identification was done using a combination of the Na and Ca CFA signals together with absolute time markers (e. g. volcanic eruptions). The variations in the higher resolution Ca data were used to complement the Na data. More details can be found in the respective paper.*

I suggest changing the statement of line 238 ff. "This indicates that the LA-ICP-MS technique is capable of identifying annual signals in this depth of the core, which would not be visible in the Na signal of the CFA alone." Based on Figure 6, Na is the most plausible indicator of annual signals, and the LA-ICP-MS agrees well with CFA.

*Yes, in general it agrees well but still shows some higher frequency variations compared to the CFA data (e. g. between 83.45 and 83.6 m). We will therefore refrain from changing that statement.*

Minor and some technical comments that were skipped in the first round of review

Line 51: Reinhardt et al. (2003) used an IR laser, not UV.

*Added IR in Line 52 rev. manuscript*

Line 53: The main difference among the systems is also the design of the ablation chamber: Two volume vs. single volume, fast vs. slow washout, etc.

*Added a respective sentence L 55f revised manuscript.*

Line 55: It is unclear what is meant by "laser cell" – ablation chamber?

*Changed to ablation chamber.*

Table 2: Helium flow: Does an inner and an outer volume exist that has separate flow rates?

*No, only one helium flow is adjustable.*

Line 117: Here it is "ablation chamber" – please check for consistency

*Done.*

Line 129: "no significant sublimation" – this is another reason why the optical mosaics are important. The fact that the grain boundaries are visible indicates some sublimation has happened (after surface decontamination). I would suggest phrasing this as "no additional increase in sublimation", although I am not 100% sure if this is what the authors are trying to say, and how this would be determined – are we talking about visual observations?

*The grain boundaries are already faintly visible immediately after microtoming. They will constantly deepen over time, which is the fundamental physical process and can only be slowed, not stopped. We did however not observe a deepening of the grain boundary grooves that would have caused the laser to go out of focus, which would then lead to insufficient ablation. "No significant sublimation" refers to the sublimation of the main ice sample body, which would for example be indicated by a rounding of the outer edges. This was not observed during the measurement and on the sample pictures. We added a short remark in L 130f rev. manuscript.*

Line 155: Was the dwell time really 1s for 238U?

*Typo, corrected to 1 ms*

Line 166: Does "laser power" refer to nominal fluence settings?

*Changed to nominal laser fluence, L 171 rev. manuscript*

Line 167 – 169: this statement is unclear. What is a "drift in laser system sensitivity"?

*This was related to the need of readjustment of the fluence settings and was misleading. We removed the sentence and reformulated it (L 170-174 rev. manuscript)*

Line 177: "Therefore the acquisition time is the dominating factor regarding depth resolution compared to the much smaller washout time." See comment above. The washout time for 150μm spot size is unknown.

*See comment above, paragraph changed.*

Line 184: Remove "therefore" in this sentence.

*Done.*

Line 223: I think there is a "for" missing here: "… for which … "

*For added.*

Table 4 is a very good overview. Was the rectangular spot not used? If it was used, it should be included here.

*It was only used for the ice section from the Last Interglacial, which is no longer discussed in the revised version of the paper.*

Line 315: Is "500 year old" correct?

*Yes.*

*Hoffmann, H. M., Grieman, M. M., King, A. C. F., Epifanio, J. A., Martin, K., Vladimirova, D., Pryer, H. V., Doyle, E., Schmidt, A., Humby, J. D., Rowell, I. F., Nehrbass-Ahles, C., Thomas, E. R., Mulvaney, R., and Wolff, E. W.: The ST22 chronology for the Skytrain Ice Rise ice core – Part 1: A stratigraphic chronology of the last 2000 years, Clim. Past, 18, 1831–1847, https://doi.org/10.5194/cp-18-1831-2022, 2022.*

Jerše, A., Mervič, K., van Elteren, J. T., Šelih, V. S., & Šala, M. (2022). Quantification anomalies in single pulse LA-ICP-MS analysis associated with laser fluence and beam size. Analyst, 147(23), 5293-5299. https://doi.org/10.1039/D2AN01172G